# Growth of electroautotrophic microorganisms using hydrovoltaic energy through natural water evaporation

Guoping Ren[1], Jie Ye [1], Qichang Hu [1], Dong Zhang [1], Yong Yuan [2] ✉ & Shungui Zhou [1] ✉

It has been previously shown that devices based on microbial biofilms can generate hydrovoltaic energy from water evaporation. However, the potential of hydrovoltaic energy as an energy source for microbial growth has remained unexplored. Here, we show that the electroautotrophic bacterium *Rhodopseudomonas palustris* can directly utilize evaporation-induced hydrovoltaic electrons for growth within biofilms through extracellular electron uptake, with a strong reliance on carbon fixation coupled with nitrate reduction. We obtained similar results with two other electroautotrophic bacterial species. Although the energy conversion efficiency for microbial growth based on hydrovoltaic energy is low compared to other processes such as photosynthesis, we hypothesize that hydrovoltaic energy may potentially contribute to microbial survival and growth in energy-limited environments, given the ubiquity of microbial biofilms and water evaporation conditions.

Phototrophy and chemotrophy represent two widely recognized forms of microbial metabolism[1–3]. Phototrophs, including algae and cyanobacteria, utilize sunlight to convert carbon dioxide and water into energy-rich organic compounds through photosynthesis[4–6]. In contrast, chemotrophs harness their energy by breaking down various organic and inorganic compounds via enzymatic reactions[2,7]. Certain microorganisms possess dual metabolic capabilities, manifesting phototrophic and chemotrophic modes of metabolism[8,9]. This metabolic versatility grants them the adaptability to thrive in diverse environments, contributing significantly to ecological diversity and resilience. However, microbial communities inhabiting extreme environments, characterized by high temperatures, high salinity, extreme acidity or alkalinity, anaerobic conditions, extreme aridity, and even the early Earth, encounter challenges in accessing conventional energy sources due to the lack of photosynthetic structures and limited availability of electron donors[7,10]. Consequently, there is an increasing interest in exploring alternative energy sources capable of sustaining microbial life within these challenging environments.

Electroactive microorganisms (EAMs) have attracted significant interest for their ability to directly exchange electrons with solid materials in the past few decades. This includes both donating to and uptaking electrons from solid materials. Various microorganisms, such as bacteria, eukaryotes, and archaea, have been extensively studied for their electron exchange capabilities[11–14]. The discovery of EAMs has unveiled new pathways in microbial metabolism and energy acquisition, driven by their unique ability to uptake extracellular electrons. For instance, Lu et al.[1] demonstrated that non-phototrophic chemoautotrophic and heterotrophic EAMs like *Alcaligenes faecalis* and *Acidithiobacillus ferrooxidans* could directly uptake extracellular electrons released from the photocatalysis of metal sulfides and oxides to support their growth[1]. In addition, Yang et al.[15] reported that EAMs such as the nonphotosynthetic $CO_2$-reducing bacterium *Moorella thermoacetica* can uptake extracellular electrons from cadmium sulfide nanoparticles to sustain cellular metabolism under light exposure[15]. More recently, a discovery by Yamamoto et al.[16] elucidated the ability of electroautotrophic microorganisms to thrive on

[1]Fujian Provincial Key Laboratory of Soil Environmental Health and Regulation, College of Resources and Environment, Fujian Agriculture and Forestry University, Fuzhou, China. [2]Guangdong Key Laboratory of Environmental Catalysis and Health Risk Control, School of Environmental Science and Engineering, Institute of Environmental Health and Pollution Control, Guangdong University of Technology, Guangzhou, China. ✉e-mail: yyuan2017@gdut.edu.cn; sgzhou@fafu.edu.cn

extracellular geoelectrical electrons in deep-sea hydrothermal environments[16]. These findings underscore the potential of naturally occurring extracellular electrons within microbial surroundings as a viable alternative energy source for cellular growth and metabolism. This offers a compelling supplement to the reliance on light or redox reactions between chemicals.

Water evaporation is a widespread and natural physical process that occurs within the global water cycle[5,17]. It amounts to approximately $5.2 \times 10^{11} m^3/yr$, which is 57 times greater than the global water supply available to mankind[18,19]. The process involves heat transport and dynamic mass phenomena, constituting a significant energy flow[20]. In this context, a revolutionary hydrovoltaic energy generation process was proposed to harness the interaction between water molecules and hydrophilic groups of artificial materials during the natural process of water evaporation, which converted ambient heat energy into sustainable hydrovoltaic electrical energy[21–23]. Water evaporation through the utilization of hydrophilic materials, such as carbon, $Al_2O_3$, and silicon nanowires, has shown potential for generating sustainable hydrovoltaic energy[22,24,25]. Recent advancements have developed this concept by investigating the hydrovoltaic process of natural biomaterials[26,27], aiming to explore the presence of this distinct process within natural environments. Remarkably, it was observed that hydrovoltaic energy generated from water evaporation through microbial biofilms could achieve voltages of ~ 0.4 V with currents around 5 μA[28–30]. This electrical energy output surpasses the minimum energy requirements for microbial growth, generally around 0.1 V and 0.02 μA[1,8,11]. These revelations open new avenues for exploring the potential interactions between ubiquitous water evaporation and microorganisms, particularly in the realm of energy conversion and metabolism. Yet, the feasibility of directly harnessing water evaporation-induced hydrovoltaic electrons (WE-HE) within microbial biofilms as a novel energy source to sustain and promote microbial growth remains a frontier to be fully explored and delineated.

Here, we present experimental evidence for a previously unreported microbial metabolism process. This process involves the generation of hydrovoltaic electrons through natural water evaporation in a biofilm, which not only supports the survival of microorganisms but also enables their growth through extracellular electron uptake. To prove this concept, we selected a typical electroautotrophy strain[8], *Rhodopseudomonas palustris*, to construct the hydrovoltaic biofilm. In addition, we examined other microorganisms including electroautotrophic and typical heterotrophic EAM strains, as well as non-EAM strains, for comparative purposes. The growth pattern of *R. palustris* was dependent on the intensity of water evaporation, correlating with sustainable carbon ($CO_2$) fixation and nitrate ($NO_3^-$) reduction. Stable-isotope probing and transcriptomic analyses revealed that $CO_2$ fixation into cellular biomass relied on the WE-HE. Our calculations about the efficiency of water evaporation-to-cellular biomass conversion indicated that the WE-HE pathway was less efficient than traditional photosynthesis, but it represents a potential source of energy due to the widespread occurrence of water evaporation and its substantial energy potential.

## Results
### Microbial growth powered by hydrovoltaic energy
To explore the potential of water evaporation-induced hydrovoltaic energy as an energy source for microbial metabolism, we first examined the autotrophic growth of *R. palustris* in a biofilm. *R. palustris* is a typical autophototrophy that can fix $CO_2$ coupled $NO_3^-$ reduction by uptaking extracellular electrons under dark, anoxic conditions[9]. To establish the optimal growth conditions, a $CO_2$ purged, anaerobically autotrophic medium supplemented with $NO_3^-$ and ascorbic acid. Notably, under dark and anoxic conditions, no significant growth of *R. palustris* was observed in the absence of extracellular electrons supply

when only ascorbic acid was present (Supplementary Fig. 1). This finding supports the notion that ascorbic acid does not function as a carbon or energy source for the growth of *R. palustris*. According to previous studies[1,31,32], ascorbic acid was used as an antioxidant to protect the cells in this system. The viability of the cells to thrive under these specific conditions would suggest the potential occurrence of a process wherein extracellular electrons, generated from the biofilm during water evaporation, enable $CO_2$ fixation and $NO_3^-$ reduction (Fig. 1a, b). To create the biofilm, *R. palustris* cells were attached to a solid electrode that was supported by the filter membrane, allowing it to float on the water column (Fig. 1a and Supplementary Fig. 2a, b). The biofilm was then arranged in a sandwich structure by overlaying another electrode. The purpose of utilizing these electrodes was to establish an external electrical circuit for the sustained generation of the WE-HE within the biofilm[1,5]. To assess the effectiveness of this microbial hydrovoltaic energy generation (M-HEG) configuration, the generation of the WE-HE from this M-HEG biofilm was quantified by recording the electric current produced in the biofilm. Scanning electron microscopy (SEM) images confirmed that *R. palustris* cells evenly distributed as a biofilm on the electrode (Supplementary Fig. 2a). Moreover, confocal laser scanning microscopy (CLSM) indicated an initially high cell live/dead ratio of 16/1 (Supplementary Fig. 2c).

To validate the contribution of hydrovoltaic energy to microbial growth, a series of studies were undertaken by manipulating the conditions of water evaporation and electrode connection (Fig. 1c and Supplementary Fig. 3a–c). In the absence of water evaporation, the measured cell biomass, quantified in terms of total cell proteins, exhibited a decrease of approximately 20%, likely due to catabolic processes in a nutrition-limited environment[15]. The high-density microbial cells could not access external energy sources to sustain their cellular metabolism, which led them to rely on their biomass by triggering the death and lysis of a portion of the cells[33,34]. Meanwhile, only a marginal increase in biomass was observed in scenarios where water evaporation occurred but without electrode connection (open-circuit mode). However, under conditions where both water evaporation and electrode connection were present, a substantial 82% increase in cell biomass for *R. palustris* cells was observed within 50 days (Fig. 1c). The growth of cells in the biofilm exhibited a strong positive linear correlation ($R^2 = 0.975$) with the charges generated from the biofilm, indicating a link between cell growth and WE-HE. CLSM images revealed a higher live/dead ratio (approximately 7:1) in the biofilm in the presence of water evaporation compared to the absence of water evaporation (approximately 1:10) (Fig. 1d). Structural analysis of the biofilm showed a spatial distribution of live and dead cells, featuring thicker total and living cell layers in the presence of water evaporation compared to its absence (Fig. 1d, Supplementary Fig. 3d and Supplementary Movie 1). Moreover, the biomass, as indicated by biofilm weight and total DNA, also increased, providing further evidence of *R. palustris* growth powered by hydrovoltaic energy (Supplementary Fig. 3e, f). In addition, the generation of WE-HE in the biofilm was regulated by gradually changing factors that influence water evaporation, such as wind velocity, relative humidity, and evaporation area ratio, to further elucidate the correlation between the WE-HE and cell growth. An increase in current intensities was observed with higher water evaporation intensities (Supplementary Fig. 4a). As expected, microbial growth, in terms of biomass accumulation, exhibited a positive correlation with current intensity (Supplementary Fig. 4b), highlighting the significance of the WE-HE as an energy source for supporting microbial growth. Conversely, in a fully sealed non-evaporating biofilm, neither current generation nor cell growth was observed (Figs. 1c and 2a), further reinforcing the conclusion that the WE-HE serves as the sole energy source for microbial growth in this system.

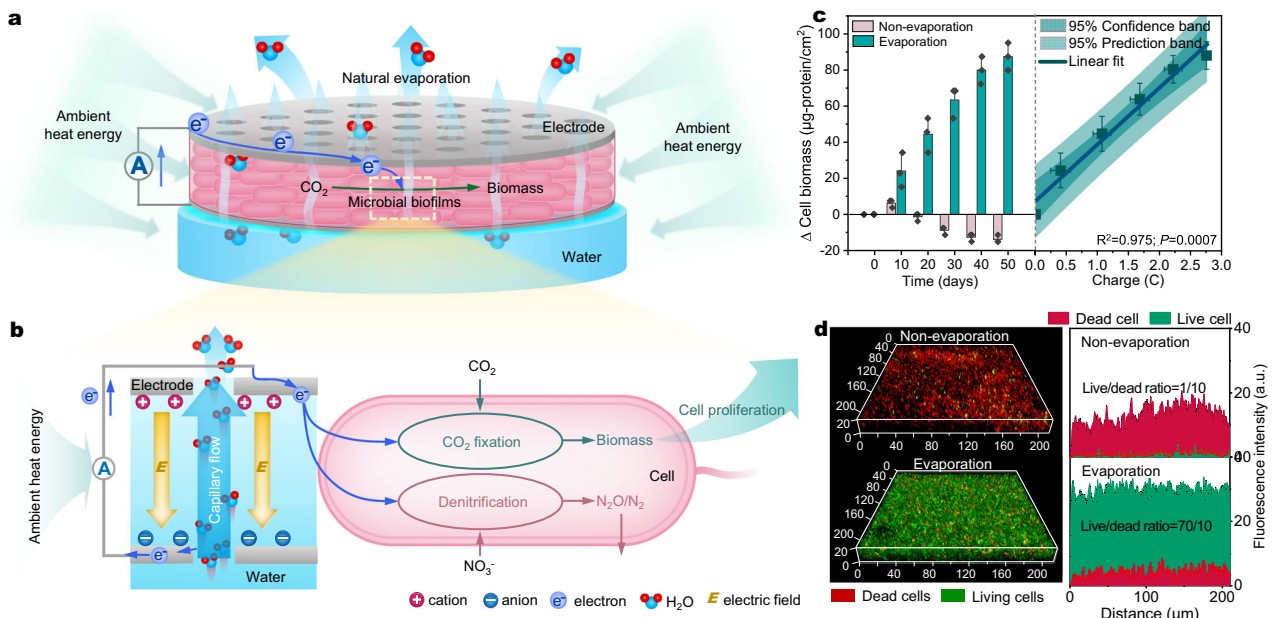

**Fig. 1 | Microbial growth powered by hydrovoltaic energy. a** Schematic diagram of natural water evaporation-enabled hydrovoltaic electron generation within biofilm from ambient heat energy for microbial growth. **b** Mechanistic illustration of the *R. palustris* metabolism by uptaking WE-HE. **c** Cell biomass (total protein) accumulation under hydrovoltaic effect and negative control (non-evaporation and electric connection), and positive relationships between cell biomass accumulation and generated charge quantity (hydrovoltaic electrons) by natural water evaporation in the *R. palustris* biofilms. Data represent mean ± SD from *n* = 3 technical replicates from one experiment. Statistical analysis was conducted with paired two-tailed *t* tests. All *P* values are provided in the Source Data. **d** Three-dimensional confocal laser scanning microscopy (CLSM) images of the *R. palustris* biofilms at 50 days, and the live/dead cell ratio of the biofilms. Scale bars: the unit is μm in (**d**).

## Generation of water evaporation-enabled hydrovoltaic energy through microbial biofilms

The *R. palustris* biofilm produced a sustainable short-circuit current of 0.52-0.87 μA and an open-circuit voltage of 0.29-0.34 V for at least 50 days through water evaporation (Fig. 2a and Supplementary Fig. 5a). Conversely, negligible short-circuit current and open-circuit voltage were generated from the biofilm in the absence of water evaporation, indicating a direct reliance on this process for energy generation. The energy generation of the biofilm was further confirmed by a switching-polarity test (Supplementary Fig. 5b), which revealed a change in induced voltage direction when the electric circuit connection was reversed without altering the biofilm's construction or position. Furthermore, the generated hydrovoltaic energy was able to produce a high load current (0.25-0.50 μA) to charge commercial capacitances. This observation indicated that the WE-HE could be directed externally to power electrical devices (Supplementary Fig. 5c–e). Kelvin probe force microscopy (KPFM) illustrated significant variations in surface potential between the top and bottom of the biofilm (Fig. 2b), indicative of distinct charge states. This difference in surface potential is positively correlated with the open-circuit voltage of the hydrovoltaic devices[35]. The *R. palustris*-based biofilm was found to possess a notable pore structure and enrichment in surface hydrophilic groups, such as -COOH, -OH, -NH$_2$, and -NH$_3^+$ (Supplementary Fig. 6). These findings align with previous identifications highlighting the pivotal role of these groups in influencing the performance of the M-HEG biofilm for the generation of WE-HE[36,37].

The generation of WE-HE was likely started from the capillary transport of water in the biofilm. The spontaneous evaporation of water evaporation leads to the formation of a bottom-up water gradient within the biofilm, resulting in the upward capillary transport of water molecules (Fig. 2c)[23,29]. As water molecules flow in the capillary channels of biofilms, the negatively charged nanochannels in the biofilms repel OH$^-$ ions while allowing the passage of H$^+$ ions in an evaporation-driven water flow. This process establishes a streaming potential and charge accumulation along the flow, creating an electric field (Fig. 2d)[38,39]. Consequently, a diffusion

current ($I_{diffuse}$) is produced, flowing in the opposite direction to the streaming current ($I_{sc}$) due to coulomb forces. Once $I_{sc}$ and $I_{diffuse}$ reach dynamic equilibrium, as indicated by $|I_{sc}| = |I_{diffuse}|$, the device stabilizes with a numerically stable open circuit voltage ($V_{oc}$)[24,40], resulting in the top electrode being positive (Fig. 2e). Upon connection to an external circuit, the electrons repelled from the bottom electrode will migrate towards the top electrode due to the presence of an electric field. The electroactive microorganisms located near the cathode (top electrode) uptake these electrons for their growth and metabolism (Fig. 2f). Thermal imaging demonstrated a decrease in the surface temperature of the biofilm due to water evaporation (Supplementary Fig. 7), suggesting potential conversion of ambient heat energy into electrical energy within the biofilm[17,18,41].

## Mechanisms of hydrovoltaic electron-driven microbial growth

In culture medium, *R. palustris* could grow by directly taking extracellular electrons through coupling CO$_2$ fixation and NO$_3^-$ reduction as the electron acceptors under dark and anoxic conditions[9]. Specifically, the hydrovoltaic electrons were able to support the growth of *R. palustris* by utilizing only CO$_2$ as the electron acceptor in the absence of NO$_3^-$ (Supplementary Fig. 3c). To validate the CO$_2$ fixation process by *R. palustris* in the M-HEG biofilm, a stable isotope assay with labeled $^{13}$C-NaHCO$_3$ was conducted. In addition, the NO$_3^-$ reduction process was also monitored to confirm its' role as an electron acceptor for cell growth. As shown in Fig. 3a, Time-of-flight secondary-ion mass spectrometry (ToF-SIMS) images of the biofilms at 50 days showed that $^{13}$C signals were distributed in the biofilm operated underwater evaporation condition, but no obvious $^{13}$C signals were observed from the biofilm without water evaporation process. A level of δ$^{13}$C 1623‰ was observed in the M-HEG biofilm underwater evaporation condition, while the negative controls (single non-evaporation and single open circuit model) showed no incorporation of $^{13}$C labeled carbon in the biofilms (Fig. 3b). This result verified the hypothesis that *R. palustris* cells used inorganic C as the carbon source via autotrophic CO$_2$ fixation for growth during water evaporation.

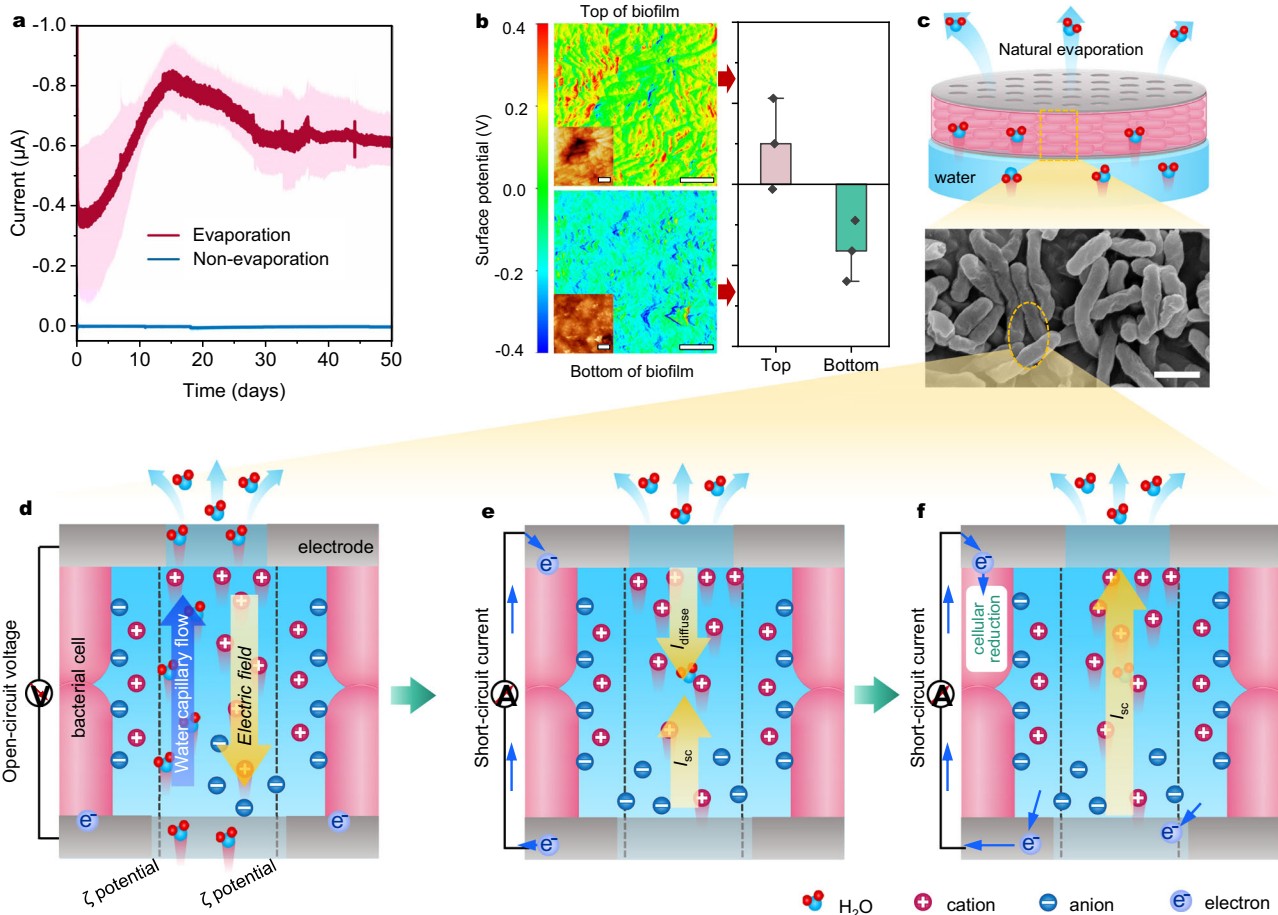

**Fig. 2 | The generation of WE-HE through microbial biofilms. a** Current profile generated by the *R. palustris* biofilms through evaporation and non-evaporation of water. The shaded area represents one standard deviation. **b** Relative surface potential maps of the *R. palustris* biofilm during water evaporation (the inset is the atomic force microscopy (AFM) topography image). **c** Schematic diagram of the electricity generation induced by evaporation in the biofilm and the scanning electron microscopy (SEM) image of the *R. palustris* biofilm. **d** Spontaneous evaporation of water in the negatively charged nanochannels in the biofilm. **e** The diffusion current ($I_{diffuse}$) and the opposing streaming current ($I_{sc}$) arised from ionic gradient and coulomb forces, respectively. **f** Electroactive microorganisms uptake extracellular hydrovoltaic electrons. Data represent mean ± SD from $n = 3$ technical replicates. Scale bars: 1 μm in (**c**).

Notably, the efficiency of $NO_3^-$ reduction with water evaporation was increased linearly as a function of hydrovoltaic electron generation (Fig. 3c and Supplementary Fig. 8a), potentially attributed to the continuous and stable energy supply supporting the metabolism of *R. palustris* biofilm. Remarkably, minimal current generation and reduction of $NO_3^-$ were observed in the absence of the *R. palustris* biofilm (Supplementary Fig. 8b, c), highlighting the critical role of the biofilm in these processes. Under conditions of water evaporation, the *R. palustris* biofilm displayed complete and continuous $NO_3^-$ reduction (-98%, $P < 0.001$), producing $N_2O$ and $N_2$ as the main nitrogenous products (Fig. 3c and Supplementary Fig. 8d). The $NO_3^-$ concentration decreased by 16% in the absence of water evaporation (Fig. 3c), possibly due to the presence of residual biosynthetic intermediates or electron donors during the cultivation process of *R. palustris*[15]. The intermittent evaporation process, consisting of 12-hours cycles to mimic day-night cycles, induced a trapezoidal reduction performance of $NO_3^-$ (Supplementary Fig. 8e). This finding strongly suggests that water evaporation is a critical factor in the biological reduction of $NO_3^-$. Taken together, we propose a novel microbial growth mode characterized by the maintenance of cell growth and metabolism through the electron uptake derived from a robust water evaporation process. The interplay among water evaporation, $CO_2$ fixation, and $NO_3^-$ reduction is crucial in sustaining *R. palustris* cell growth and metabolic processes.

Transcriptomics of the biofilm was performed to further verify the cell growth by coupling carbon fixation and denitrification processes in the M-HEG biofilm. The score plots of principal component analysis (PCA) showed that genes from *R. palustris* biofilms with and without evaporation were clustered into different groups (Fig. 4a). A comparison between *R. palustris* biofilms with and without water evaporation revealed that 1329 genes were upregulated while 587 genes were downregulated (fold change (FC) > 2, $P < 0.05$), as shown in Fig. 4b. These genes encode various metabolic processes, such as electron transfer activity, ATPase activity and biosynthetic process (Supplementary Fig. 9a). Both flagella and nanowires were important components for extracellular electron uptake by *R. palustris*[9]. Remarkably, the transcriptional levels of genes encoding these components ($P < 0.05$) were significantly elevated in the biofilm with water evaporation (Fig. 4c), indicating a stimulated potential electron transfer process mediated by these two components under this condition. In addition, *R. palustris* cell might also uptake extracellular electrons via c-type cytochromes (Cyt *c*) because those genes encoding Cyt *c* were also upgraded with water evaporation (Fig. 4c). In particular, a significant upregulation of gene encoding cytochrome $c_2$ (Cyt $c_2$) was observed, which could enable transferring electron inward through the periplasm to complex III for ubiquinone cycling[9,42], generating reduction equivalents by reducing NAD(P)$^+$ to NAD(P)H and adenosine diphosphate (ADP) to adenosine triphosphatase (ATP) (Fig. 4c and

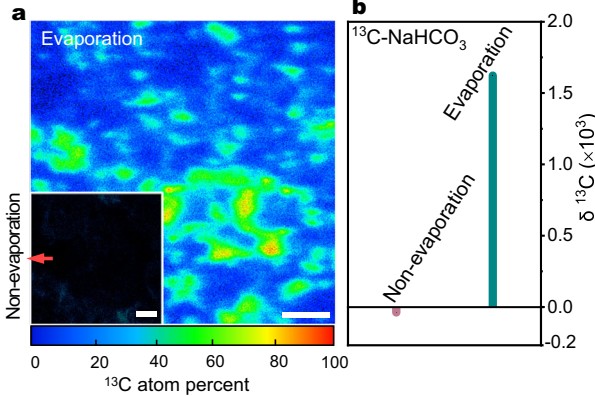

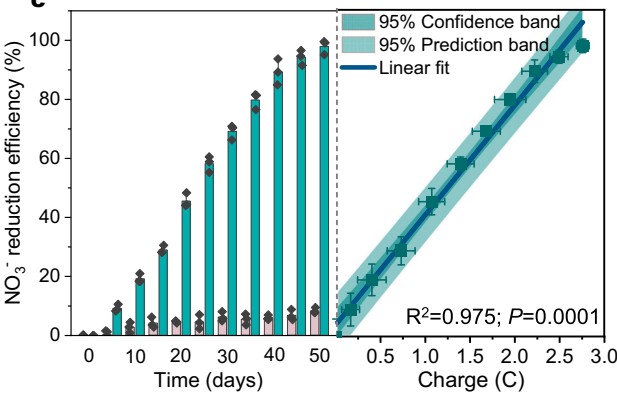

**Fig. 3 | Microbial metabolism of *R. palustris* powered by hydrovoltaic energy.** **a** Time-of-flight secondary-ion mass spectrometry (ToF-SIMS) images of the biofilms at 50 days with labeled $^{13}C$-NaHCO$_3$ fixation, the inset is the biofilms without water evaporation. **b** Labeled $^{13}C$-NaHCO$_3$ fixation by the biofilms at 50 days. **c** Typical time course of NO$_3^-$ reduction by *R. palustris* under hydrovoltaic effect and deletional control (non-evaporation), and positive relationships between NO$_3^-$ reduction efficiency and generated charge quantity (hydrovoltaic electrons) by natural water evaporation. Data represent mean ± SD from $n = 3$ technical replicates from one experiment. Statistical analysis was conducted with paired two-tailed $t$ tests. Scale bars: 4 μm in (**a**) and in the inset image.

Supplementary Fig. 9b, c). Moreover, the denitrification enzymes (nitrate reductase, nitrite reductase, nitric oxide reductase, and nitrous oxide reductase) were significantly upregulated (2.5-4.5-fold, 2.1-24.2-fold, 6.1-fold and 2.1-39.4-fold, respectively) with water evaporation compared to the absent control (Fig. 4c). To confirm electron transfer function of these highly expressed components for *R. palustris* metabolism, inhibition experiments were performed by using carbonyl cyanide m-chlorophenyl hydrazine (CCCP, an inhibitor of the formation of nanofilaments via the dissipation of proton motive force) and antimycin A (an inhibitor of cytochrome $bc_1$)[8,43]. A decline in NO$_3^-$ reduction performance was observed upon the addition of either inhibitor (Supplementary Fig. 9d, e), indicating the involvement of these components in the electron transfer process for NO$_3^-$ reduction. It is well known that a Calvin-Benson-Bassham (CBB) cycle was involved in CO$_2$ fixation for *R. palustris*[8,9]. As expected, the CBB cycle was significantly activated by hydrovoltaic energy ($P < 0.05$) compared to the absent control (Fig. 4c). Specifically, the key genes encoding those enzymes including ribulose-1,5-bisphosphate carboxylase/oxygenase (RuBisCO, -5.8-fold, $P < 0.05$), glyceraldehyde 3-phosphate dehydrogenase (*gapdh*, -5.0-fold, $P < 0.05$) and phosphoribulokinase (*prk*, -9.5-fold, $P < 0.05$), were upregulated significantly (Supplementary Fig. 10). These findings, alongside labeled $^{13}C$-CO$_2$ fixation and NO$_3^-$ reduction, suggested that hydrovoltaic electrons could sufficiently energize microbial metabolisms.

## Utilization of hydrovoltaic energy by diverse electroactive microorganisms

The growth powered by hydrovoltaic energy in other microorganisms was investigated as well, including other autotrophic EAMs, typical heterotrophic EAMs, and non-EAMs. Two well-known other electro-autotrophic bacteria, *Thiobacillus denitrificans* and *M. thermoacetica*, demonstrated the capability to use extracellular electrons for cell growth[13,15]. Their biofilms, underwater evaporation conditions, displayed evident growth, resulting in a growth ratio of 86% and 89%, respectively, as well as notably high cellular viabilities (Fig. 5a, b and Supplementary Fig. 11a, b). In contrast, two typical heterotrophic EAMs (*Shewanella oneidensis* and *Geobacter sulfurreducens*)[44,45] and two non-electroactive bacteria (*Escherichia coli* and *Bacillus subtilis*)[46] exhibited almost no noticeable growth (Fig. 5c–f). *G. sulfurreducens* and *S. oneidensis* are typical electroactive bacteria known for their ability to transfer electrons generated by oxidizing organic compounds to the extracellular environment, leading to electricity generation[47,48]. In addition, they can directly utilize extracellular photoelectrons and electrode electrons to drive reduction reactions. Despite facing restrictions in growth due to the absence of preferred carbon sources in the medium[49,50], these heterotrophic electroactive microorganisms (EAMs) were able to maintain a well-functioning metabolic state by directly taking up and utilizing hydrovoltaic electrons (Fig. 5d). Specifically, *S. oneidensis* and *G. sulfurreducens* displayed microbial metabolisms by reducing NO$_3^-$ and methyl orange[51,52], respectively, utilizing the hydrovoltaic electrons generated by the biofilms (Supplementary Fig. 11c, d). Furthermore, a significant positive correlation was observed between the accumulation of hydrovoltaic electrons and the microbial growth and metabolism processes in the EAMs (Supplementary Fig. 11b–d), with $R^2$ values ranging from 0.953 to 0.994. On the contrary, no cellular growth and low cellular viability were observed in the *E. coli-* and *B. subtilis*-based biofilms (Fig. 5e), which must be due to their incapacity to receive extracellular electrons. These results validate the ability of electroautotrophic microorganisms to grow and metabolize by harnessing the WE-HE through uptaking extracellular electrons.

## Discussion

Water evaporation is a widespread occurrence on Earth, occurring on various surfaces such as rivers, lakes, wetlands, soils, oceans, and even plant leaves[17,53]. Microbial biofilms, a form of collective life with emergent properties and a much higher level of organization than single microbial cells, dominate various Earth habitats, accounting for 40-80% of bacterial and archaeal cells ($1.2 \times 10^{30}$ cells)[54,55]. These biofilms inherently align with water evaporation[56,57]. There is speculation about the possibility of the widespread occurrence of hydrovoltaic energy generation through water evaporation from biofilms. It was found that the generated electrons effectively supported biomass synthesis in microorganisms with a considerable energy conversion efficiency. Over 50 days, 30.99 kJ of ambient heat energy ($\Delta E_w$) was absorbed through water evaporation, generating 0.22 J of electric energy (termed as hydrovoltaic energy, $\Delta E_e$). This corresponds to an energy conversion efficiency of $7.1 \times 10^{-3}$ ‰ from ambient heat energy to hydrovoltaic energy (Fig. 5g and Supplementary Table 1). Impressively, 96% of this hydrovoltaic energy could be further converted into chemical energy ($\Delta E_b$) for biosynthesis in microorganisms (Fig. 5g and Supplementary Table 2). However, by considering the energy flux of each sub-process, the overall energy conversion efficiency for microbial growth was found to be $6.7 \times 10^{-3}$ ‰. It is worth noting that the overall energy conversion efficiency was primarily limited by the process of converting ambient heat energy to hydrovoltaic energy through water evaporation. This particular step relies heavily on the utilization of the ambient heat energy for the phase transition of water from a liquid to a gas state[17]. Although natural water evaporation for microbial growth displays lower conversion efficiency compared to

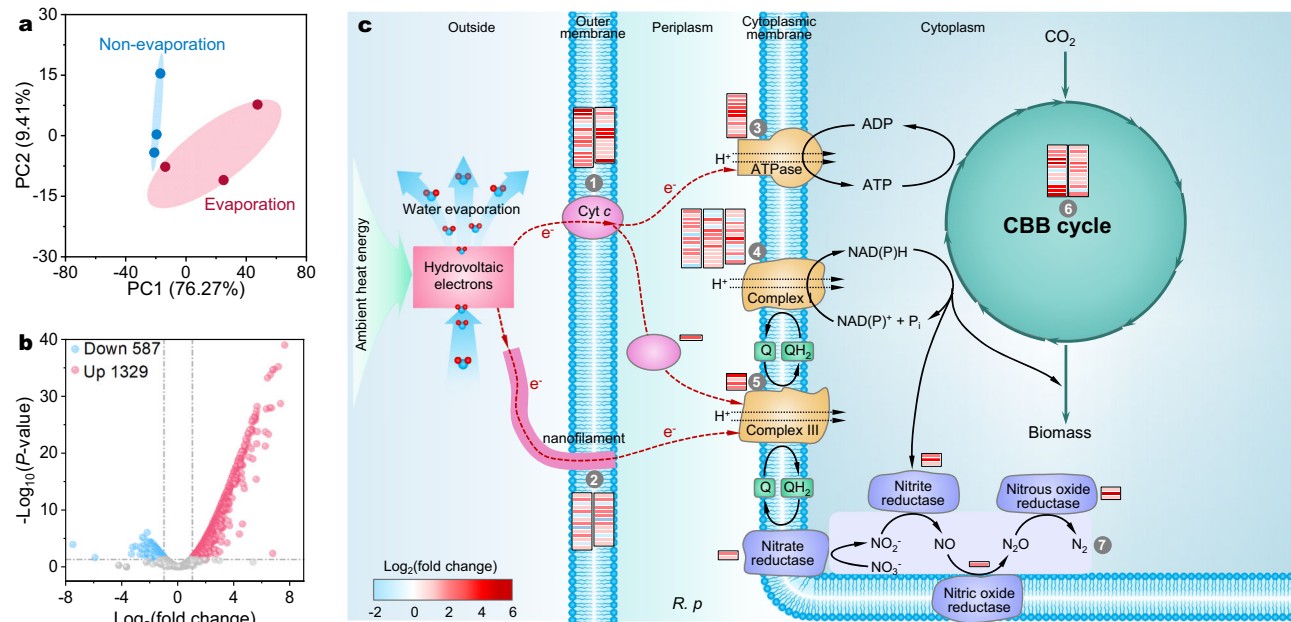

**Fig. 4 | Transcriptomic analyses of *R. palustris* cells.** Principal component analysis (PCA, 95% confidence) (**a**) and differential gene volcano graph (**b**) of the *R. palustris* biofilms powered by the WE-HE as compared with that under non-evaporation. Dashed lines represent the gating of fold change (FC) > 2.0 and $P < 0.05$. **c** Hypothetical model of electron and carbon flow driven by the WE-HE: (1) *C*-type cytochromes, (2) Nanofilaments of *R. palustris* such as protein nanowires and flagella, (3) ATP synthase, (4) NADH dehydrogenase complex, (5) ubiquinol: *c*-type cytochrome oxidoreductase complex, (6) Calvin-Benson-Bassham (CBB) cycle, (7) enzymes for denitrification, e.g., nitrate reductase, nitrite reductase, nitric oxide reductase and nitrous oxide reductase. Statistical analysis was conducted with paired two-tailed *t*-tests, three biological replicates per treatment ($n = 3$). All $P$ values are provided in the Source Data.

the other natural process such as natural photocatalysis based on light-induced and mineral-mediated pathways ($13 \times 10^{-2}$ ‰) and natural photosynthesis (1%)[1,5,15], it remains a plentiful and widely available energy source due to the prevalence of microbial biofilms and water evaporation conditions.

This research demonstrates that hydrovoltaic electrons can function as an energy source for driving microbial growth through natural water evaporation, presenting a novel mechanism. Although some initial mechanisms have been proposed to elucidate the origins of electrons and protons in biofilms through water evaporation, further research is necessary to thoroughly investigate these sources in future studies. Particularly, in laboratory conditions, artificial conductive metal wires are utilized to establish an external electrical circuit that aids in the movement of electrons from the lower to the upper layers of the biofilm. Interestingly, natural environments host microbial nanowires and conductive minerals present, suggesting the potential for naturally occurring self-contained electrical pathways within biofilms[58,59], enhancing electron transfer in the biofilm. Hydrovoltaic electrons are continuously delivered to EAMs for intracellular carbon fixation and the reduction of compounds such as $NO_3^-$, and methyl orange. Microorganisms are abundantly found in various ancient environments[60]. It is speculated that, during the early stages of anoxic Earth, hydrovoltaic electrons may have served as an energy source, sustaining microbial growth. This hypothesis provides a new means of interpreting the evolution and early development of life. In modern environments, microorganisms have been also discovered in extreme settings, ranging from deserts to subterranean rivers[10,61]. These locales often lack organic matter, typically produced by chemolithotrophs. However, the ubiquitous process of water evaporation at the water-solid interface generates hydrovoltaic electrons. This phenomenon may directly or indirectly facilitate microbial metabolism and support the growth of other microorganisms, such as heterotrophs and chemoautotrophs. This study advances our understanding of the potential impact of water evaporation on biological processes and energy cycling in the natural world.

In conclusion, this study investigated the utilization of hydrovoltaic electrons generated through water evaporation in microbial biofilms as an energy source for microbial growth. The research primarily focused on *R. palustris* as a model organism and examined its growth patterns and metabolic processes in the presence of the WE-HE. The results demonstrated that the WE-HE effectively supported the cellular growth of *R. palustris* by coupling the $CO_2$ fixation and nitrate reduction processes. Furthermore, the study explored the potential of other microorganisms to utilize hydrovoltaic energy through extracellular electron uptake, noting that other EAMs were able to harness hydrovoltaic electrons for growth and/or metabolism. Overall, the findings highlight the prospective role of hydrovoltaic energy as an alternative and sustainable energy source within microbial ecosystems, unveiling a novel mechanism elucidating how microbial growth and energy cycles operate within natural environments.

## Methods
### Bacterial strain and growth protocol
The *R. palustris* strain CGMCC 1.2180 was obtained from the China General Microbiological Culture Collection Center (CGMCC). To facilitate anaerobic photoheterotrophic growth, *R. palustris* was cultivated in heterotrophic Medium I (Supplementary Table 3), under continuous illumination at a temperature of 30 °C and an optical power density of 25 W/m²[9]. When the growth of *R. palustris* reached the logarithmic phase, the *R. palustris* cells were sequentially centrifuged, washed, and resuspended with 0.9% NaCl solution three times for preparing concentrated cell suspensions. In all cases requiring a change of medium, microbial cells were washed three times with 0.9% saline after centrifugation at $7000 \times g$. Subsequently, 200 µL of cell suspensions (15 mg/mL, dry weight) were evenly coated on a titanium mesh to form a biofilm. The same approach was employed for cultivating other bacterial biofilms. The *G. sulfurreducens* PCA strain (ATCC 51573) was obtained from the American Type Culture Collection (ATCC) and grown in heterotrophic Medium I and experimented in autotrophic Medium II at 30 °C (Supplementary Table 4). The *S.*

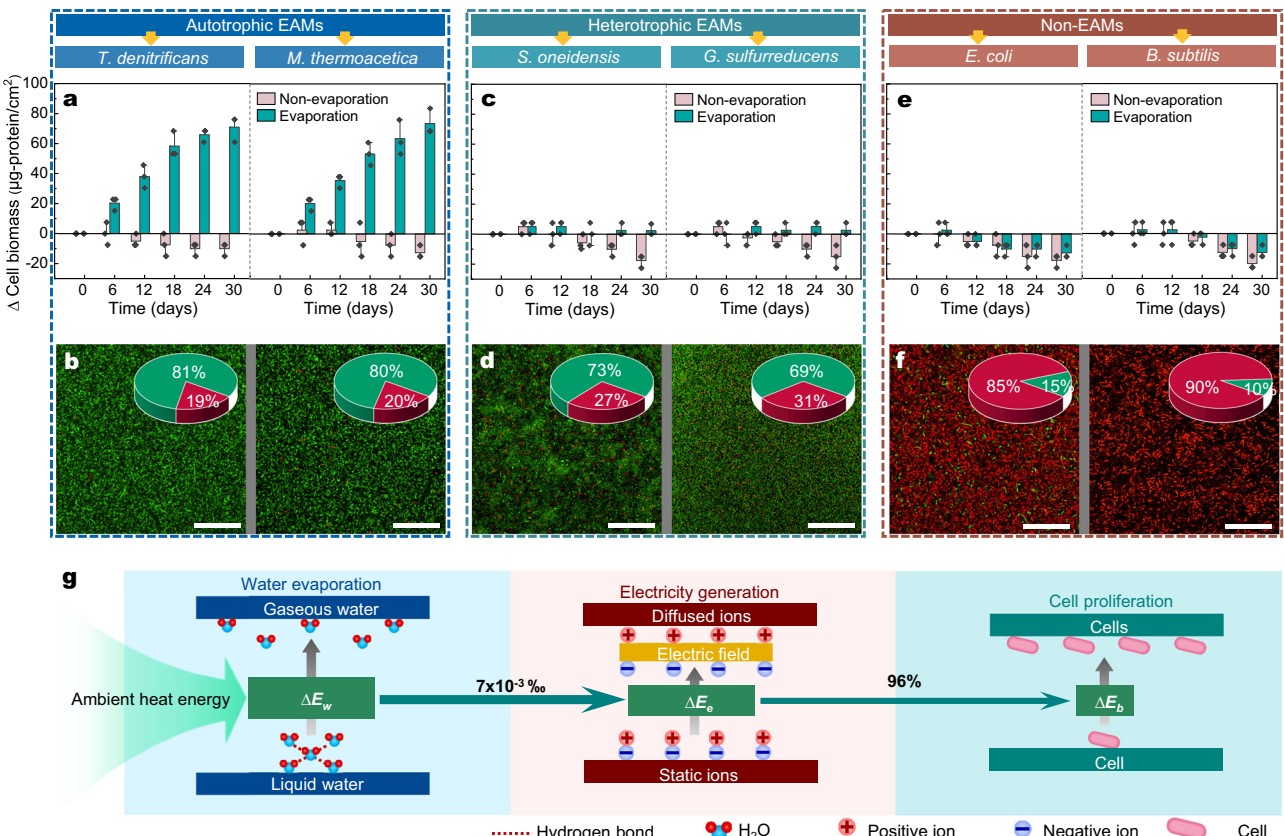

**Fig. 5 | Hydrovoltaic energy utilized by different microorganisms. a, b** Cell biomass accumulation (**a**) and CLSM images (**b**) with autotrophic EAMs (*T. denitrificans* and *M. thermoacetica*). The pie graph represents the live/dead cell ratio of the biofilms (red and green represent dead and live cells, respectively). **c, d** Cell biomass accumulation (**c**) and CLSM images (**d**) with heterotrophic EAMs (*S. oneidensis* and *G. sulfurreducens*). **e, f** Cell biomass accumulation (**e**) and CLSM images (**f**) with non-EAMs (*E. coli* and *B. subtilis*). **g** Overall energy conversion from ambient energy to chemical energy in microbial cells, calculation details refer to the Supplementary Information. $\Delta E_w$ is the absorbed heat energy by water evaporation, $\Delta E_e$ is the generated electric energy by water evaporation, $\Delta E_b$ is the energy required for cell proliferation (including biomass increase and $NO_3^-$ reduction). Data represent mean ± SD from $n = 3$ technical replicates from one experiment. Scale bars: 40 μm in (**b**), (**d**), and (**f**).

*oneidensis* (ATCC 700550), *B. subtilis* (ATCC 6051) and *E. coli* (BAA-3219) strains were acquired from the ATCC and incubated in heterotrophic Medium I (LB medium) and experimented in autotrophic Medium II at 30 °C, 30 °C and 37 °C, respectively (Supplementary Tables 5, 6). *M. thermoacetica* (ATCC39073) was purchased from ATCC, and cultured in heterotrophic Medium I, and experimented in autotrophic Medium II at 52 °C (Supplementary Table 7). *T. denitrificans* (DSM 12475) was purchased from the Leibniz Institute DSMZ-German Collection of Microorganisms and Cell Cultures, and cultured in in heterotrophic Medium I and experimented in autotrophic Medium II at 30 °C (Supplementary Table 8).

### Fabrication of M-HEG system

To fabricate the M-HEG system, the obtained biofilms were placed between a top electrode and a bottom electrode using porous titanium mesh electrodes. This sandwiched setup was then inserted into a transparent vessel and supported by the filter membrane. A fresh autotrophic Medium II was added to a glass tube, providing sufficient water and minerals for water evaporation and cellular growth. To ensure a pure culture, the entire experimental procedure was conducted under sterile conditions. Firstly, the equipment, nozzles, syringes, and culture media required for the preparation of M-HEG were subjected to autoclave sterilizing. Subsequently, a 200 μL suspension of pure cells (15 mg/mL) was used to prepare the biofilm, which was then assembled into the M-HEG system. The fabricated system was placed in a constant temperature chamber at 30 °C, maintaining a dark and anaerobic environment with a $CO_2/N_2$ ratio of 20%/80%. The top

and down titanium mesh electrodes were linked to the electrical circuit through titanium wires. To enable optimal electron flow, a short circuit was created by directly connecting the top and down electrodes with titanium wires, without external resistance loading on the electrical circuit (Supplementary Fig. 2) To maintain low relative humidity (10-20% RH) for stable and continuous water evaporation (evaporation group), sterilized and dried allochroic silica gel (Sinopharm Chemical Reagent Co., Ltd., China) was used as a drying agent. The system was sealed with airtight tape to prevent water-gas exchange, resulting in an evaporation loss of 2.7 ± 0.7‰. This setup served as the control for absent water evaporation (non-evaporation group). Each trial was performed with a minimum of three replicates unless explicitly stated otherwise.

### Characterizations of biofilm

To visualize the structure of the biofilms, a scanning electron microscope (SEM, SU8020, Hitachi, Japan) was employed to observe the biofilm and filter membrane structures. Assessing the surface properties, contact angle measurements were performed using the Krüss DSA 10 Mk2 system (Germany). Temperature and infrared heat images were captured by the FLIR infrared heat imager (USA). The functional groups present were analyzed using Fourier transform infrared (FTIR) spectroscopy and X-ray photoelectron spectroscopy (both by Thermo Fisher, USA). Pore structure analysis was conducted using an automatic specific surface and porosity analyzer (Mack Instruments Inc., USA). The viability of *R. palustris* biofilms during water evaporation was examined using confocal laser scanning microscopy (LSM880,

ZEISS, Germany). The resultant fluorescence data were analyzed using ZEN software and Image J software. Before observation, the biofilms were stained with the Live/Dead BacLight Bacterial Viability Kit (Invitrogen, CA).

### Electrical measurements

The electricity measurements were conducted using electrical measurement systems (4200A-SCS, Keithley, USA; PalmSens4, PALMSENS, Netherlands). Before performing voltage and current tests, the circuit parameters were set to zero for both current and voltage. In particular, the load current and voltage were measured across various resistances. Commercial capacitors were acquired from an online marketplace (Taobao, China) and charged using the hydrovoltaic system. The output power density was calculated by multiplying the load current by the load voltage. The ion conductivity and zeta potential were determined using a conductivity meter (DDS-307A, China) and a Zetasizer Nano ZS (Malvern, UK) respectively. To obtain the electrostatic potential map, a relative surface potentials map was measured using atomic force microscopy (AFM) with a Kelvin probe (Kelvin probe force microscopy, KPFM) on an Atomic Force Microscope (Dimension Icon, Bruker, Germany).

### Evaluation of microbial growth and metabolism functions

To assess biomass variation during water evaporation, the biofilm was collected and washed at least three times. The resulting biofilm cells were dehydrated using a vacuum freeze dryer (Alpha 1–4 LDplus, Christ, Germany) at −55 °C for 24 h. The dehydrated biofilm was weighed using a precision balance (ME54, Mettler Toledo, China) to determine the change in biofilm weight. The total DNA of the biofilm was extracted using the TIANamp Bacteria DNA Kit (TIAN-GEN, China) and quantified using the NanoDrop 2000 spectrophotometer (Thermo Fisher Scientific, USA). The total protein (as a measure of cell biomass) was extracted using the One-step Bacterial Active Protein Extraction Kit (Sangon Biotech Co., Ltd., China) and quantified using a multimode reader (SpectraMax iD3, Molecular Devices, USA) with the PierceTM BCA Protein Assay Kit (Thermo Scientific, USA). The NADH/NAD$^+$ ratio was determined using the NAD$^+$/NADH Quantification Kit (Millipore Sigma, USA) and a multimode reader. ATP concentration was determined using the ATP Assay Kit (Beyotime Biotechnology, China) by measuring chemiluminescence with the multimode reader. Inhibition experiments were conducted by adding different inhibitors during the CES process, including antimycin A (Maokangbio, China) and carbonyl cyanide m-chlorophenyl hydrazine (CCCP, Macklin, China)[1]. Stock solutions of antimycin A and CCCP were prepared in 100% DMSO and stored as aliquots at −20 °C for immediate use. Methyl orange decolorization was analyzed using a spectrophotometer (UV2600, Sunny Hengping Instrument).

The concentrations of $NO_3^-$ and $NO_2^-$ were analyzed using an ion chromatograph (ICS 900, Thermo Fisher Scientific, USA). To detect the gas products ($N_2$ and $N_2O$) produced during the denitrification process, the M-HEG was assembled into a sealed blue-capped bottle with a butyl-rubber stopper, in which some amount of dried allochroic silica gel was used to maintain a lower environmental humidity and facilitate the water evaporation process (Supplementary Fig. 12). The concentration of $N_2O$ in the headspace was sampled and analyzed using a gas chromatograph (Agilent 7890, Agilent Technologies, USA) equipped with an electron capture detector (ECD), while the concentration of $N_2$ was analyzed using a thermal conductivity detector (TCD).

### Stable isotope probing and SIMS analyses

To validate the microbial assimilation of $CO_2$ by the *R. palustris* biofilm during water evaporation, $^{13}C$-labeled $NaHCO_3$ (99 atom % $^{13}C$) was utilized as the sole carbon source in the autotrophic culture medium. Over 50 days of subjecting the biofilm to water evaporation, the cells

were collected through centrifugation and subsequently dehydrated by freeze-drying. The dried cells (5 mg) were then analyzed for $CO_2$ fixation using a stable isotope ratio mass spectrometer (Isoprime 100, Elementar, Germany). ToF-SIMS analysis was performed using a Time-of-Flight Secondary-ion mass spectrometry instrument (IONTOF TOF-SIMS5, Germany). A specific region underwent analysis twice, with the introduction of $^{13}C$-labeled biofilms during the second analysis, specifically aimed at confirming the presence of $^{13}C$ in the biofilms. To ensure precision, individual ion image frames were aligned using the $^{13}C$ ion, and the epifluorescence microscopy image was transformed to match the ToF-SIMS ion image[2].

### Transcriptomics

Triplicate biofilm cells were collected and preserved at −80 °C using RNAlater (Invitrogen, USA). Total RNA was extracted using TRIzol reagent (Invitrogen, USA). For library preparation, the NEBNext Ultra II Directional RNA Library Prep Kit was used directionally. Biotinylated rRNA probes were employed to facilitate the removal of rRNA through subtractive hybridization. The resulting mRNA was sequenced on Illumina HiSeq/MiSeq platforms. The raw sequencing data underwent quality checking and filtering. Reads matching 16 S and 23 S RNA genes were discarded, and the remaining reads were aligned against the published genome of *R. palustris* CGMCC 1.2180 (NZ_CP058907.1). The mapped reads were normalized using FPKM (fragments per kilobase per million mapped reads).

### Reporting summary

Further information on research design is available in the Nature Portfolio Reporting Summary linked to this article.

## Data availability

The data supporting the findings of this study are available within the paper and its supplementary information. The RNA-seq data generated in this study have been deposited in the NCBI Trace Archive database under accession code PRJNA1110727. Source data are provided with this paper.

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

## Acknowledgements

This work was supported by the National Science Fund for Distinguished Young Scholars (41925028, S.Z.), the National Natural Science Foundation of China (42307176, G.R.), and the Guangdong Basic and Applied Basic Research Foundation (2023B1515040022, Y.Y.).

## Author contributions

G.R., Y.Y., and S.Z. conceived the idea for this work. G.R. and Q.H. cultured the microbial strains. G.R., J.Y., and D.Z. performed the characterizations, catalytic measurements, and transcriptomic experiments. G.R., Y.Y., and J.Y. analyzed the data. G.R., Y.Y., and S.Z. wrote the manuscript. All the authors contributed to the interpretation of the data and preparation of the manuscript.

## Competing interests

The authors declare no competing interests.
