## [Peer Review File · Nature Communications]

Growth of electroautotrophic microorganisms using hydrovoltaic energy through natural water evaporationReviewer #1 (Remarks to the Author):

In this manuscript, the possibility of the use of hydrovoltaic energy as an energy source for microbial growth and metabolism was investigated. The phenomenon of electricity production from the hydrovoltaic energy generated from water evaporation through microbial biofilms has been reported earlier. However, whether this energy can support microbial growth and metabolism remains to be thoroughly investigated. This study demonstrates that the hydrovoltaic energy generated from water evaporation supports the growth and metabolism of *Rhodopseudomonas palustris* and a few other bacteria under autotrophic conditions through a series of experiments. Based on these findings, a distinct energy conservation process in microbes other than phototrophy and chemotrophy is invoked. My specific comments on different aspects of the study are as follows.

- Lines 100-101: Supplementary Table 1: Can ascorbic acid added in a reasonably high concentration (0.3 g/L) serve as the carbon and energy source in the "autotrophic medium"? A control experiment with a medium lacking ascorbic acid may be required to confirm or rule out its role and carbon/energy source in the here-tested experimental conditions.
- Results section "Microbial growth powered by hydrovoltaic energy and generation of water evaporation enabled...."
 - Fig 1: How the non-evaporation condition of growing *Rhodopseudomonas palustris* biofilm is maintained while it is designed by sealing the upper lid with tape.
 - An abiotic control is missing, i.e., to check hydrovoltaic energy generation and nitrate reduction without microbial biofilm. Also, a control experiment with the medium lacking nitrate is required to confirm that the microbes used no other electron acceptor in the growth process.
- Results section "Generation of water evaporation-enabled hydrovoltaic energy through microbial biofilms"
 - An abiotic control experiment without microbial biofilm is required to confirm or rule out the role of abiotic reactions/processes in generating hydrovoltaic energy in the here-tested experimental conditions.
- Results section: Utilization of Hydrovoltaic energy by diverse EAM.
 - Lines 236-238: The growth of heterotrophic EAMs may be restricted due to the lack of the required/preferred carbon sources in the medium (E.g., Acetate for *Geobacter sulfurreducens* and Lactate for *S. oneidensis*). Hence, they might have shown limited growth and activity.
 - Lines 241-242: The substrates used in the medium also serve as the source of electrons in this case. How are the authors sure these microbes used only the hydrovoltaic electrons generated by the biofilms for the respective (nitrate and methyl orange) reduction activities?
 - Lines 246-248: The claim should focus more on the hydrovoltaic energy use by autotrophic EAMs rather than heterotrophic EAMs since there are other possible electron donors in the growth medium used in the case of heterotrophic EAMs.
- Lines 280-281: This statement requires further studies and evidence in natural settings/microcosm experimental setups.
- Lines 69-71: References regarding natural biomaterial should be referred to here.
- Lines 310-311: What was the cell density in 200 μ L cell suspension? How was uniformity maintained/ensured in the inoculum size in all experiments?
- How was the sterility maintained for experimental (M-HEG) setups? How exactly does nutritional supply from medium to microbes occur in these setups?
- Lines 375-376 (and Lines 185-186): How were N₂ and N₂O gases monitored in these systems? Also, mention the nitrate reduction products that were analyzed with the ion chromatograph.
- Supplementary information Table 2: In equation ($P = I \times V / 4$), what is the factor 4?
- Supplementary information Figure 7b: What are the "other" denitrification products? How come nitrate is the denitrification product, and why is it shown in this graph?
- The title can be made more specific by replacing "electroactive" by "electroautotrophic" microorganisms.

Reviewer #2 (Remarks to the Author):

The manuscript discusses the potential of hydrovoltaic energy generated by water evaporation, which has not been explored as a microbial growth energy source. It identifies a new microbial

metabolic pathway that connects hydrovoltaic energy from water evaporation to microbial metabolism. The study shows that water evaporation-induced hydrovoltaic electrons effectively support the growth of certain bacteria, emphasizing sustainable carbon fixation and nitrate reduction. This microbial survival mechanism, reliant on hydrovoltaic energy, might be common in various microorganisms capable of extracellular electron uptake. This pathway sheds light on a previously overlooked energy source for microorganisms in low-energy environments, differing from traditional phototrophy and chemotrophy.

Thank you for the opportunity to review this interesting study. The data portraying cell growth (e.g., Fig.1C) are compelling. However, I'm uncertain about the origin of electrons that facilitate cellular growth in the autotrophic medium. Hydrovoltaic processes can generate a sustainable voltage potential between the "wet" and "dry" regions of the hydrovoltaic apparatus, resulting in charge separation and an electron flow through the electrical circuit. Charge recombination occurs without a net production or consumption of electrons.

In simpler terms, while hydrovoltaic processes can generate a driving force, the process of promoting cellular growth (such as reducing CO₂ into organic metabolites) requires electron consumption. I'm interested to know the source of these essential electrons for growth. Without this clarification, I am unable to recommend the publication of this manuscript in its current form.

Reviewer #3 (Remarks to the Author):

The authors have proposed an interesting and novel idea to grow electroactive microbes via hydrovoltaic energy. While the idea is commendable, the manuscript does not sufficiently address the underlying mechanisms. My specific comments to the authors are given below:

1. Line 116-117: The authors must make it clear how non-evaporation of water increases catabolic metabolism? What is the underlying reason? Similarly, how does water evaporation enhance anabolic metabolism?
2. In the setup with the biofilm and electrodes, the authors must describe how current generation is possible in the first place. Do they use a specific set of resistors to make electrons flow? Or apply a potential? It is highly unclear how electricity is generated in this setup. A clear schematic diagram along with a picture of the setup must be provided, instead of multiple images in the text and supplementary.
3. It is surprising to note that the current obtained from the setup is negative (Fig 2a), while the corresponding supplemental figure 4 shows a positive voltage. Can the authors explain why it is so? Are the electrons actually flowing from the biofilm to the electrode as stated in the text? A negative current implies that the biofilm may be uptaking electrons from the electrodes. A clear explanation is needed to address this anomaly.
4. One of my concerns is that water evaporation may not provide sufficient activation energy for the biofilm to generate electricity. Could the authors show that the thermodynamics of water evaporation favours energy generation? I understand this is an additional step, but it is necessary to show that the thermodynamics actually favour the reaction pathway proposed here. We understand the thermodynamics pretty well for conventional microbial fuel cells so it must also be accounted for here.
5. Lines 163-167: How big is the potential difference across the biofilm? Is it sufficiently high for sustained generation of current over long periods of time? And can this potential difference be maintained over time?
6. Even for the non-evaporation setup where tape is used as a sealant, there is likely to be some evaporation, though it is minimised. The authors must account for this.
7. Lines 237-238: *Geobacter* and *Shewanella* exhibited no growth. Despite this, the authors claim

that these microbes maintained a well-functioning metabolic state by actually uptaking hydrovoltaic electrons. How is this possible? And how are *Geobacter* and *Shewanella* able to reduce MO and NO_3^- in the absence of an applied potential? These statements are not satisfactory, and need more evidence and explanation.

Reviewer #4 (Remarks to the Author):

I have completed my review of the manuscript titled "Growth of electroactive microorganisms using hydrovoltaic energy through natural water evaporation" by Ren et al. and would like to provide my feedback and recommendations.

Overall, the manuscript presents an interesting study that investigates the coupling of electron generation by hydrovoltaic energy with the microbial metabolism of the anoxygenic phototrophic bacterium *Rhodospseudomonas palustris*. The authors aim to determine if *R. palustris* can utilize electrons generated from water evaporation within a hydrovoltaic device to drive its chemoautotrophic metabolism coupled with nitrate reduction. The experimental approach is appropriate, and the results are well presented and support the conclusions and claims made by the authors.

However, I do have a concern regarding the novelty of this work. It has been previously demonstrated by the authors (Liu et al. *Sci. Adv* 7, eabh1852, 2021) that *R. palustris* is capable of accepting electrons directly from solid surfaces, as well as from another microorganisms through a syntrophic relationship. In the present work, the authors have replaced the source of electrons from being the electrode of a bioelectrochemical system to a hydrovoltaic energy device. The authors suggest that a similar mechanism involving heme-containing proteins is responsible for electron uptake, as also shown in the above-cited paper. Furthermore, the integration of hydrovoltaic devices with electroactive organisms has been previously reported (e.g., Liu et al, *Nat Commun* 13, 4368, 2022), including by the authors themselves (e.g., Ren et al, *Chem. Eng. J.* 441, 135921, 2022).

In light of these previous studies, I believe it is crucial for the authors to elaborate further on how the present work adds novel elements to the existing literature. While I acknowledge the excellent quality of this work, it is important to ensure that the manuscript clearly highlights its novelty compared to the previously published studies before being considered for publication in *Nature Communications*.

Reponses to Reviewer Comments for:

“Growth of electroautotrophic microorganisms using hydrovoltaic energy through natural water evaporation”

Reviewer #1 (Remarks to the Author):

*[1] In this manuscript, the possibility of the use of hydrovoltaic energy as an energy source for microbial growth and metabolism was investigated. The phenomenon of electricity production from the hydrovoltaic energy generated from water evaporation through microbial biofilms has been reported earlier. However, whether this energy can support microbial growth and metabolism remains to be thoroughly investigated. This study demonstrates that the hydrovoltaic energy generated from water evaporation supports the growth and metabolism of *Rhodospseudomonas palustris* and a few other bacteria under autotrophic conditions through a series of experiments. Based on these findings, a distinct energy conservation process in microbes other than phototrophy and chemotrophy is invoked. My specific comments on different aspects of the study are as follows.*

Response: Thank you for your insightful comments that are helpful for improving our manuscript.

[2] Lines 100-101: Supplementary Table 1: Can ascorbic acid added in a reasonably high concentration (0.3 g/L) serve as the carbon and energy source in the “autotrophic medium”? A control experiment with a medium lacking ascorbic acid may be required to confirm or rule out its role and carbon/energy source in the here-tested experimental conditions.

Response: We express our sincere gratitude to the reviewer for their valuable suggestions. Given that *R. palustris* is typically considered an autotrophic strain, it should not utilize ascorbic acid as a carbon and energy source. To exclude the possibility of ascorbic acid serving as a carbon and energy source, *R. palustris* was anaerobically incubated in the dark without the provision of exogenous electron donors while in the presence of 0.3 g/L ascorbic acid. Under these culture conditions, no significant increase in OD₆₀₀ or total protein content was observed (**Supplementary Fig. 1**), thus experimentally confirming that ascorbic acid cannot support the growth of *R. palustris* as a carbon and energy source.

Ascorbic acid has been widely utilized as an antioxidant in electrochemical and photochemical systems (*Nat. Commun.* 3, 768, 2012; *Sci. Adv.* 7, eabe2631, 2021; *Nat. Nanotechnol.* 5, 73-79, 2010). In our previous study (*ISME J*, 17, 163-171, 2023), we observed that radical production is specific to *R. palustris* during their extracellular electron-accepting energy metabolism in dark and anoxic environments. Therefore, ascorbic acid was employed as an antioxidant to protect the cells in this system.

Revised in text: “To establish the optimal growth conditions, a CO₂ purged, anaerobically autotrophic medium supplemented with NO₃⁻ and ascorbic acid. Notably, under dark and anoxic conditions, no significant growth of *R. palustris* was observed in the absence of extracellular electrons supply when only ascorbic acid was present (**Supplementary Fig. 1**). This finding supports the notion that ascorbic acid does not function as a carbon or energy source for the growth of *R. palustris*. According to previous studies^{1,31,32}, ascorbic acid was used as an antioxidant to protect the cells in this system.” **Please see Page 5 Line 96-101.**

Revised in the Supplementary information: Following new figures have been added into the Supplementary information.

Supplementary Fig. 1 | Effect of ascorbic acid on the growth of *R. palustris*. **a**, OD₆₀₀ of *R. palustris* incubated in the presence of ascorbic acid under a dark and anaerobic environment. **b**, Cell biomass (total protein) accumulation of *R. palustris* in the presence of ascorbic acid under a dark and anaerobic environment. Data represent mean ± SD from *n* = 3 technical replicates from one experiment.

References

- R1. Lu, A. et al. Growth of non-phototrophic microorganisms using solar energy through mineral photocatalysis. *Nat. Commun.* **3**, 768 (2012).
- R2. Swainsbury, D. J. et al. Structures of *Rhodospseudomonas palustris* RC-LH1 complexes with open or closed quinone channels. *Sci. Adv.* **7**, eabe2631 (2021).
- R3. Iwuchukwu, I. J. et al. Self-organized photosynthetic nanoparticle for cell-free hydrogen production. *Nat. Nanotechnol.* **5**, 73-79 (2010).
- R4. Huang, L. et al. Light-independent anaerobic microbial oxidation of manganese driven by an electrosynthetic coculture. *ISME J.* **17**, 163-171 (2023).

[3] Fig 1: How the non-evaporation condition of growing *Rhodospseudomonas palustris* biofilm is maintained while it is designed by sealing the upper lid with tape.

Response: Thanks for your question. To maintain a non-evaporation condition, it was essential to prevent contact between the *R. palustris* biofilm and the gas environment. In this regard, we covered the biofilm with a layer of highly viscous non-permeable tape (duct tape, Deli, China) to prevent the water evaporation. To achieve optimal prevention of water evaporation, the tape was in close proximity

to the biofilm to prevent potential water evaporation induced by the gap between the tape and the biofilm. In addition, the tape was wrapped around the edges of the biofilm to prevent water molecules from escaping into the air environment through any gaps, thereby preventing water evaporation. We evaluated the evaporation under sealed conditions, resulting in an evaporation of 0.037 ± 0.01 g over 50-day period. The amount of water evaporation in the non-evaporation setup was only $2.7 \pm 0.7\%$ of that in the evaporation setup, indicating successful achievement of a non-evaporation control.

Revised in text: “The system was sealed with airtight tape to prevent water-gas exchange, resulting in an evaporation loss of $2.7 \pm 0.7\%$. This setup served as the control for absent water evaporation (non-evaporation group).” Please see Page 13-14 Line 338-340.

[4] An abiotic control is missing, i.e., to check hydrovoltaic energy generation and nitrate reduction without microbial biofilm. Also, a control experiment with the medium lacking nitrate is required to confirm that the microbes used no other electron acceptor in the growth process.

Response: Thank you for the reviewer's comments. Following the suggestions, we performed two additional control experiments: one with the medium lacking nitrate and another with the abiotic control without *R. palustris* biofilm. The outcomes of these experiments are now depicted in **Supplementary Fig. 3c** and **Supplementary Figs. 8b** and **c** in the revised manuscript. We found that the hydrovoltaic electrons were able to support the growth of *R. palustris* by utilizing only CO₂ as the electron acceptor in the absence of NO₃⁻ (**Supplementary Fig. 3c**). Remarkably, the system lacking the *R. palustris* biofilm only produced a current of approximately 5 nA and showed no reduction of nitrate (**Supplementary Figs. 8b, c**), indicating its limited ability to produce energy.

A control experiment without nitrate exhibited no significant impact on the growth of *R. palustris* (**Supplementary Fig. 3c**), suggesting the ability of *R. palustris* to utilize CO₂ in an autotrophic medium as an electron acceptor for growth through the uptake of hydrovoltaic electrons. Notably, *R. palustris* has the capacity to utilize both CO₂ and nitrate as electron acceptors concurrently (*Sci. Adv.* 7, eabh1852, 2021). In our investigation, nitrate reduction served as an additional indicator of the metabolic activity of microbial cells driven by hydrovoltaic electrons.

Revised in text: “Specifically, the hydrovoltaic electrons were able to support the growth of *R. palustris* by utilizing only CO₂ as the electron acceptor in the absence of NO₃⁻ (**Supplementary Fig. 3c**).” Please see Page 8 Line 177-179.

“Remarkably, minimal current generation and reduction of NO₃⁻ were observed in the absence of the *R. palustris* biofilm (**Supplementary Figs. 8b, c**), highlighting the critical role of the biofilm in these processes.” Please see Page 8 Line 191-193.

Revised in the Supplementary information: The following new figures have been added into the Supplementary information

Supplementary Fig. 3c, Cell biomass (total protein) accumulation of control experiment by removing nitrate in an autotrophic Medium II.

Supplementary Fig. 8 c, Typical time course of NO_3^- reduction performance in the absence of hydrovoltaic effect (dead *R. palustris* biofilm and without *R. palustris* biofilm). **d**, Hydrovoltaic current of abiotic control (without *R. palustris* biofilm).

References

R1. Liu, X. et al. Syntrophic interspecies electron transfer drives carbon fixation and growth by *Rhodospseudomonas palustris* under dark, anoxic conditions. *Sci. Adv.* **7**, eabh1852 (2021).

[5] *An abiotic control experiment without microbial biofilm is required to confirm or rule out the role of abiotic reactions/processes in generating hydrovoltaic energy in the here-tested experimental conditions.*

Response: Thanks. As suggested by the reviewer, the abiotic control experiment without microbial biofilm has been conducted and described in **Question [4]**.

[6] *Lines 236-238: The growth of heterotrophic EAMs may be restricted due to the lack of the required/preferred carbon sources in the medium (E.g., Acetate for *Geobacter sulfurreducens* and Lactate for *S. oneidensis*). Hence, they might have shown limited growth and activity.*

Response: We greatly appreciate the reviewer's comments. As you pointed out, *Geobacter sulfurreducens* and *Shewanella oneidensis* are typical heterotrophic microorganisms (*Water Res.* **208**, 117860, 2022; *Chem. Eng. J.* **471**, 144727, 2023). They require specific organic carbon sources (e.g., acetate for *G. sulfurreducens* and lactate for *S. oneidensis* (*Environ. Sci. Technol.* **45**, 815-820, 2011; *P. Natl. Acad. Sci. USA* **106**, 2874-2879, 2009) for normal growth and proliferation. We have referenced

relevant literatures in the manuscript to explain this.

Revised in text: “Despite facing restrictions in growth due to the absence of preferred carbon sources in the medium^{49,50}, these heterotrophic electroactive microorganisms (EAMs) were able to maintain a well-functioning metabolic state by directly taking up and utilizing hydrovoltaic electrons (**Fig. 5d**).”.
Please see Page 10 Line 250-252.

References

- R1. Jing, X. et al. Anode respiration-dependent biological nitrogen fixation by *Geobacter sulfurreducens*. *Water Res.* **208**, 117860 (2022).
- R2. Li, H. et al. The iron cycling mediated by a single strain *Shewanella oneidensis* MR-1 and its implication for nitrogen removal. *Chem. Eng. J.* **471**, 144727 (2023).
- R3. Geelhoed, J. S. & Stams, A. J. Electricity-assisted biological hydrogen production from acetate by *Geobacter sulfurreducens*. *Environ. Sci. Technol.* **45**, 815-820 (2011).
- R4. Pinchuk, G. E. et al. Genomic reconstruction of *Shewanella oneidensis* MR-1 metabolism reveals a previously uncharacterized machinery for lactate utilization. *P. Natl. Acad. Sci. USA* **106**, 2874-2879 (2009).

[7] Lines 241-242: The substrates used in the medium also serve as the source of electrons in this case. How are the authors sure these microbes used only the hydrovoltaic electrons generated by the biofilms for the respective (nitrate and methyl orange) reduction activities?

Response: We apologize for not providing a detailed description of the culture mediums used in the manuscript, which led to your confusion. In the hydrovoltaic electron-driven microbial reduction experiments involving water evaporation, we employed inorganic culture media (devoid of easily utilizable organic electron donors) for all microorganisms in this study. Nevertheless, the control group subjected to no water evaporation exhibited minimal reduction of nitrate and methyl orange (**Supplementary Figs. 11c, d**), indicating that the substrates in the medium were insufficient to support the microbial reduction and growth processes without the uptake of hydrovoltaic electrons. A more precise account of our initial culture medium and the medium used in the M-HEG system for different strains has now been included in the supplementary materials (**Supplementary Tables 4 to 8**).

Revised in Supplementary Information: “The *Geobacter sulfurreducens* PCA strain (ATCC 51573) was obtained from the American Type Culture Collection (ATCC) and grown in heterotrophic Medium I and experimented in autotrophic Medium II at 30 °C (**Supplementary Table 4**)¹. The *Shewanella oneidensis* (ATCC 700550), *Bacillus subtilis* (ATCC 6051) and *Escherichia coli* (BAA-3219) strains were acquired from the ATCC and incubated in heterotrophic Medium I (LB medium) and experimented in autotrophic Medium II at 30 °C, 30 °C and 37 °C, respectively (**Supplementary Tables 5, 6**)^{2,3}. *Moorella thermoacetica* (ATCC39073) was purchased from ATCC, and cultured in heterotrophic Medium I and experimented in autotrophic Medium II at 52 °C (**Supplementary Table**

7)⁴. *Thiobacillus denitrificans* (DSM 12475) was purchased from the Leibniz Institute DSMZ-German Collection of Microorganisms and Cell Cultures, and cultured in in heterotrophic Medium I and experimented in autotrophic Medium II at 30 °C (Supplementary Table 8)⁵.” Please see Page 2 Line 26-36.

[8] Lines 246-248: The claim should focus more on the hydrovoltaic energy use by autotrophic EAMs rather than heterotrophic EAMs since there are other possible electron donors in the growth medium used in the case of heterotrophic EAMs.

Response: We greatly appreciate the reviewer's suggestion. As suggested, we had modified the claim in the revised manuscript and read as follows:

Revised in text: “These results validate the ability of electroautotrophic microorganisms to grow and metabolize by harnessing the WE-HE through uptaking extracellular electrons.” Please see Page 11 Line 259-261.

[9] Lines 280-281: This statement requires further studies and evidence in natural settings/microcosm experimental setups.

Response: We appreciate the valuable suggestions provided by the reviewer. As highlighted correctly, further experimental research is required to clarify the impacts of hydrovoltaic energy on global biogeochemical cycles. In order to avoid any possible misinterpretations, we have decided to omit this statement from the revised manuscript.

In line with the reviewer's suggestion, we conducted preliminary experiments utilizing natural biofilm from a local lake to construct a microbial hydrovoltaic energy generation (M-HEG) system. We observed that electroautotrophic microorganisms could grow and metabolize driven by hydrovoltaic energy within this natural biofilm. The M-HEG with the natural biofilm produced a voltage of ~0.4 V and a current of ~0.8 μ A (Fig. R1a, b), respectively, which was comparable to those generated with the *R. palustris* biofilm. We also evaluated the nitrate reduction performance driven by this M-HEG under the water evaporation condition. As illustrated in Fig. R1c, under water evaporation, the nitrate reduction performance of the M-HEG was significantly higher than that of the non-evaporative M-HEG (Fig. R1c), indicating that hydrovoltaic energy derived from water evaporation could enhance the metabolism of natural microorganisms. Moreover, analysis of the microbial community indicated that the abundance of electroactive denitrifying microorganisms (e.g., *Clostridium_sensu_stricto_1*, *Lactococcus*, *Aeromonas*) in the M-HEG was notably higher than in the non-evaporative M-HEG (Figs. R1d, e), indicating an enrichment of these strains due to the presence of hydrovoltaic energy. Furthermore, it was observed that water evaporation significantly increased the relative abundance of the key carbon fixation functional gene (*cbbL*) (Fig. R1f), suggesting a facilitated autotrophic process

in the natural biofilm due to the presence of hydrovoltaic energy. Nevertheless, given the intricate nature of natural mixed biofilms, additional evidence is needed to confirm the potential influence of hydrovoltaic energy on biogeochemical cycling in natural ecological systems. This aspect will be the primary focus of our forthcoming research.

Fig. R1 Effect of hydrovoltaic energy on microbial community structure and metabolism activity in a natural mixed biofilm. The hydrovoltaic current (a) and voltage (b) of the M-HEG with a natural mixed biofilm under conditions with and without water evaporation. (c) NO_3^- reduction by the M-HEG under conditions with and without water evaporation. (d) Influence of hydrovoltaic energy on the abundance of electroactive (d) and denitrifying (e) microorganisms at the genus level in the biofilms. (f) Influence of hydrovoltaic energy on expression of Calvin cycle carbon fixation functional genes *cbbL* in the biofilms. Data represent mean \pm SD from $n = 3$ technical replicates from one experiment.

[10] Lines 69-71: References regarding natural biomaterial should be referred to here.

Response: As suggested by the reviewer, we have incorporated the references (*ACS Appl. Mater. Interfaces* **12**, 11232-11239, 2020; *J. Mater. Sci.* **56**, 16387-16398, 2021) into the main text.

Revised in text: “

26. Zhou, X. et al. Harvesting electricity from water evaporation through microchannels of natural wood. *ACS Appl. Mater. Interfaces* **12**, 11232-11239 (2020).

27. Zou, J. et al. Carbonization temperature dependence of hydrovoltaic conversion of natural wood. *J. Mater. Sci.* **56**, 16387-16398 (2021).

”Please see Page 18 Line 460-463.

[11] Lines 310-311: What was the cell density in 200 μL cell suspension? How was uniformity maintained/ensured in the inoculum size in all experiments?

Response: We apologize for not providing detailed process of biofilm preparation in the manuscript. The cell suspensions used in our study were all in 15 mg/mL (dry weight). In order to ensure a

consistence for all experiments, a standardized preparation process was adopted for the biofilm preparation. Firstly, the *R. palustris* strains used in the experiments were obtained at the logarithmic phase ($OD_{600} \approx 0.9$). Furthermore, to remove organic matter from the culture medium, the cells were centrifuged and washed three times. To standardize the cell concentration in each suspension, 0.5 mL of cell suspension was extracted from the total suspension for centrifugation to isolate wet bacterial cells. Subsequently, these cells were freeze-dried for 6 hours to determine their dry weight. Using this dry weight data, the 0.5 mL of cell suspension was adjusted to a concentration of 15 mg/mL (dry weight) before proceeding with the biofilm preparation. We have improved the description of the relevant experimental procedures in the revised manuscript.

Revised in Supplementary Information: “Subsequently, 200 μ L of cell suspensions (15 mg/mL, dry weight) were evenly coated on a titanium mesh to form a biofilm.” Please see Page 2 Line 25-26.

[12] *How was the sterility maintained for experimental (M-HEG) setups? How exactly does nutritional supply from medium to microbes occur in these setups?*

Response: Thanks for your question. To ensure a pure bacterial system, the entire experimental procedure was conducted under sterile conditions. Firstly, the equipments, nozzles, syringes, and culture media required for the preparation of M-HEG were subjected to high-temperature and high-pressure sterilization. They were then placed in an already sterilized anaerobic glove box ($CO_2/N_2 = 20\%/80\%$). Subsequently, within the sterile anaerobic chamber, a 200 μ L suspension of pure *R. palustris* cells (15 mg/mL) was used to prepare the biofilm, which was then assembled into the M-HEG system.

In order to provide the necessary nutritional elements for the microbial biofilm, we directly added a culture medium to the M-HEG system (as shown in the **Supplementary Fig. 2a**). The biofilm electrode was placed onto a hydrophilic filter membrane, which was floated naturally on water surface. Water with nutritional elements can be accessible to microbial cells through this filter membrane.

Revised in text: “To ensure a pure culture, the entire experimental procedure was conducted under sterile conditions. Firstly, the equipments, nozzles, syringes, and culture media required for the preparation of M-HEG were subjected to autoclave sterilizing. Subsequently, a 200 μ L suspension of pure cells (15 mg/mL) was used to prepare the biofilm, which was then assembled into the M-HEG system.”. Please see Page 13 Line 327-331.

Revised in the Supplementary information: Following figure has been modified

Supplementary Fig. 2 a, Schematic illustration of the M-HEG system. Optical microscopy image of the electrodes (top electrode and bottom electrode), scanning electron microscopy (SEM) images of the *R. palustris* biofilm and the filter membrane. Scale bars: 200 μm in **a** for top electrode, 2 μm in **a** for biofilms, 10 μm in **a** for filter membrane.

[13] Lines 375-376 (and Lines 185-186): How were N_2 and N_2O gases monitored in these systems? Also, mention the nitrate reduction products that were analyzed with the ion chromatograph.

Response: Thanks for your question. In order to detect the gas products (N_2 and N_2O) produced during the denitrification process, we used a sealed reactor as illustrated in **Supplementary Fig. 12**. The main components include a blue-capped bottle, a butyl-rubber stopper (for gas sampling), M-HEG, autotrophic culture medium, dried allochroic silica gel, and a syringe. The reactor was sterilized prior to the experiment and performed inside a sterilized anaerobic glove box. The application of the dried allochroic silica gel was to maintain a lower environmental humidity, which can facilitate the process of water evaporation. The sealed blue-capped bottle with a butyl-rubber stopper was used to collect the gas products generated during the denitrification process. Additionally, in order to obtain accurate N_2 and N_2O gas, we provided sterilized helium gas for the anaerobic glove box to eliminate the influence of N_2 and N_2O in the air. We have included the methods for each denitrification product in the **Methods** section in the revised manuscript.

Revised in Supplementary Information: “The concentrations of NO_3^- and NO_2^- were analyzed using an ion chromatograph (ICS 900, Thermo Fisher Scientific, USA). In order to detect the gas products (N_2 and N_2O) produced during the denitrification process, the M-HEG was assembled into a sealed blue-capped bottle with a butyl-rubber stopper, in which some amount of dried allochroic silica gel was used to maintain a lower environmental humidity and facilitate the water evaporation process (**Supplementary Fig. 12**). The concentration of N_2O in the headspace was sampled and analyzed using a gas chromatograph (Agilent 7890, Agilent Technologies, USA) equipped with an electron capture detector (ECD) as previously reported⁶, while the concentration of N_2 was analyzed using a thermal conductivity detector (TCD).”. Please see Page 2 Line 39-46.

Revised in the Supplementary information: “Following figure has been added into the Supplementary information.”

Supplementary Fig. 12 | Device diagram of a small-scale sealed reactor for measurement of the gas products (N_2 and N_2O) produced during the denitrification process driven by hydrovoltaic electrons.

[14] *Supplementary information Table 2: In equation ($P = I \times V / 4$), what is the factor 4?*

Response: We apologize for not including the definition of factor 4 in the description. Factor 4 is the constant used for calculating the maximum power when the hydrovoltaic I - V curve approximates a straight line. As shown in **Fig. R2**, the I - V curve indicates that the hydrovoltaic generator can generate maximum current I and voltage V . The rectangle formed by any point on the straight I - V line and the x-axis and y-axis represents the output power (P). According to the principle that the area of a rectangle is maximized when the length and width of the rectangle are half of the two perpendicular sides of a right-angled triangle respectively, the output power is maximized at $I/2$ and $V/2$, given by the equation $P = I/2 \times V/2 = I \times V / 4$ (**Formula 1**). Following the suggestions from the reviewer, we have added the definition of constant 4 in the Supplementary Information.

$$P = I \times V / 4 = I / 2 \times V / 2 \quad (1)$$

Revised in Supplementary information: “factor 4 is the constant used to calculate the maximum power when the hydrovoltaic I - V curve approximates a straight line⁸”. Please see Page 3 Line 57-59.

Fig. R2 Schematic diagram for calculating the maximum output power (P) through the I - V curve.

[15] *Supplementary information Figure 7b: What are the “other” denitrification products? How come*

nitrate is the denitrification product, and why is it shown in this graph?

Response: In order to further confirm the progress of the water evaporation-driven denitrification process ($\text{NO}_3^- \rightarrow \text{NO}_2^- \rightarrow \text{NO} \rightarrow \text{N}_2\text{O} \rightarrow \text{N}_2$), we conducted measurements on the key products of denitrification (NO_2^- , N_2O and N_2) (*Environ. Int.* **127**, 353-360, 2019). The remaining products were categorized as "other", including NO and NH_4^+ produced during the process of dissimilatory reduction of nitrate to ammonium (*Sci. Rep.*, **8**, 12769, 2018).

We apologize for the confusion caused by our inaccurate depiction in **Supplementary Fig. 7b** (shown as **Supplementary Fig 8b** in the revised Supplementary information). The original intention of this graph was to illustrate the nitrogen balance of the denitrification process, where the displayed nitrate concentration represented the remaining concentration at the final stage of denitrification. To provide a more accurate description, we have made the following modifications: changed the vertical axis to "Nitrogenous components of denitrification" and updated the caption to "Nitrogen balance of denitrification driven by hydrovoltaic electrons." These revisions have been incorporated into the supplementary materials.

Revised in the Supplementary information: Following figure has been modified

Supplementary Fig. 8b Nitrogenous components of denitrification driven by hydrovoltaic electrons at day 50.

References

- R1. Chen, M. et al. Light-driven nitrous oxide production via autotrophic denitrification by self-photosensitized *Thiobacillus denitrificans*. *Environ. Int.* **127**, 353-360 (2019).
- R2. Lo, K.-J. et al. Whole-genome sequencing and comparative analysis of two plant-associated strains of *Rhodopseudomonas palustris* (PS3 and YSC3). *Sci. Rep-UK* **8**, 12769 (2018).

[16] *The title can be made more specific by replacing "electroactive" by "electroautotrophic" microorganisms.*

Response: Thanks so much for your valuable suggestion. We have changed the term "electroactive" to "electroautotrophic" in the title in the revised manuscript.

Revised in text: "Growth of electroautotrophic microorganisms using hydrovoltaic energy through natural water evaporation". Please see Page 1 Line 1-2.

Reviewer #2 (Remarks to the Author):

[1] *The manuscript discusses the potential of hydrovoltaic energy generated by water evaporation, which has not been explored as a microbial growth energy source. It identifies a new microbial metabolic pathway that connects hydrovoltaic energy from water evaporation to microbial metabolism. The study shows that water evaporation-induced hydrovoltaic electrons effectively support the growth of certain bacteria, emphasizing sustainable carbon fixation and nitrate reduction. This microbial survival mechanism, reliant on hydrovoltaic energy, might be common in various microorganisms capable of extracellular electron uptake. This pathway sheds light on a previously overlooked energy source for microorganisms in low-energy environments, differing from traditional phototrophy and chemotrophy.*

Response: I would like to express my sincerest appreciation for the thorough and precise comments provided by the reviewer.

[2] *Thank you for the opportunity to review this interesting study. The data portraying cell growth (e.g., Fig.1C) are compelling. However, I'm uncertain about the origin of electrons that facilitate cellular growth in the autotrophic medium. Hydrovoltaic processes can generate a sustainable voltage potential between the "wet" and "dry" regions of the hydrovoltaic apparatus, resulting in charge separation and an electron flow through the electrical circuit. Charge recombination occurs without a net production or consumption of electrons. In simpler terms, while hydrovoltaic processes can generate a driving force, the process of promoting cellular growth (such as reducing CO₂ into organic metabolites) requires electron consumption. I'm interested to know the source of these essential electrons for growth. Without this clarification, I am unable to recommend the publication of this manuscript in its current form.*

Response: Thank you very much for your question. The generation of hydrovoltaic electrons involves the following processes: absorption of ambient heat energy → water evaporation → capillary transport of water → migration of ions → formation of ion concentration gradient → migration of electrons between anode and cathode → electron uptake by microorganisms". To elucidate this mechanism more clearly, relevant mechanistic illustrations have been included in **Figs. 2c-f** in the revised manuscript.

Since Guo *et al.* discovered the evaporation-driven hydrovoltaic effect in 2017 (*Nat. Nanotechnol.* **12**, 317-321, 2017), evaporation-driven electricity generation technology has garnered significant interest due to its negative carbon emissions and significant energy potential. The mechanism of energy generation in the evaporation-driven hydrovoltaic system has been extensively explored and explained in previous studies (*Nano Energy* **80**, 105522, 2021; *Nat. Nanotechnol.* **13**, 1109-1119, 2018; *Chem. Soc. Rev.* **51**, 4902-4927, 2022). The widely accepted mechanism is the classical electrokinetic effect based on the double electric layer theory. The detailed mechanism of the generation of hydrovoltaic electrons in the M-HEG can be explained as follows:

1) A double electric layer (DEL) is spontaneously formed consisting of a Stern layer and a diffusion

layer at the coulomb interface between microbial cells and water molecules. The Stern layer comprises anions that are firmly adsorbed onto the surface of microbial cells via electrostatic interactions. Cations in the diffusion layer are adsorbed close to the Stern layer owing to coulomb forces, and thus shield the electrostatic potential of the Stern layer (*Interdisciplinary Mater. I*, 449-470, 2022). Accordingly, zeta (ζ) potential emerges at the shear plane between the Stern and the diffusion layers (*Chem. Soc. Rev. 51*, 4902-4927, 2022).

2) The spontaneous evaporation of water will create a bottom-up water gradient within the biofilm, leading to capillary transport of water molecules upwards (**Fig. 2c**). When water molecules flow on a solid surface, the dynamic variations of the DEL will produce electrokinetic phenomena (**Fig. 2d**). The streaming potential is one of the classical electrokinetic effects (*Nat. Nanotechnol. 12*, 317-321, 2017).

3) When water molecules are driven by a pressure gradient through a channel with a size that equals the Debye length of solution, the narrow inner charged wall of the channel will lead to an overlap of DELs (*Interdisciplinary Mater. I*, 449-470, 2022). Due to the coulombic force, ions with opposite charges to the channel walls are filled into the channel, which reveals the selectivity of a certain ion.

4) As the water molecules continue to flow, a difference in protons concentration develops across the channel, allowing a potential difference to be created, resulting in streaming potential (ion flux), as demonstrated in **Fig. 2e**.

5) The surface of the top electrode accumulates a substantial number of positively charged cations, which subsequently attract electrons from the bottom electrode via external circuit (*Adv. Sci. 10*, 2302941, 2023; *ACS nano 17*, 18456-18469, 2023) (**Fig. 2e**). The surface of the bottom electrode, on the other hand, accumulates a large number of negatively charged anions, causing the electrons to be repelled and move away from the anions. Upon connecting to an external circuit, the electrons that are repelled from the bottom electrode will migrate towards the top electrode through the external closed-circuit (*ACS nano 17*, 18456-18469, 2023).

6) The electroactive microorganism at the top electrode will uptake these electrons for growth and metabolism via extracellular electron transfer (**Fig. 2f**).

Revised in text: “The generation of WE-HE was likely started from the capillary transport of water in the biofilm. The spontaneous evaporation of water evaporation leads to the formation of a bottom-up water gradient within the biofilm, resulting in the upward capillary transport of water molecules (**Fig. 2c**)^{23,29}. As water molecules flow in the capillary channels of biofilms, the negatively charged nanochannels in the biofilms repel OH⁻ ions while allowing the passage of H⁺ ions in an evaporation-driven water flow. This process establishes a streaming potential and charge accumulation along the flow, creating an electric field (**Fig. 2d**)^{38,39}. Consequently, a diffusion current ($I_{diffuse}$) is produced, flowing in the opposite direction to the streaming current (I_{sc}) due to coulomb forces. Once I_{sc} and $I_{diffuse}$ reach dynamic equilibrium, as indicated by $|I_{sc}|=|I_{diffuse}|$, the device stabilizes with a numerically stable

open circuit voltage (V_{oc})^{24,40}, resulting in the top electrode being positive (**Fig. 2e**). Upon connection to an external circuit, the electrons repelled from the bottom electrode will migrate towards the top electrode due to the presence of an electric field. The electroactive microorganisms located near the cathode (top electrode) uptake these electrons for their growth and metabolism (**Fig. 2f**).”. **Please see Page 7 Line 159-171.**

Revised in text: New figures describing the current generation mechanism of the M-HEG have been included in **Fig. 2** of the revised manuscript.

Fig. 2 | The generation of WE-HE through microbial biofilms. a, Current profile generated by the *R. palustris* biofilms through evaporation and non-evaporation of water. **b**, Relative surface potential maps of the *R. palustris* biofilm during water evaporation (the inset is the atomic force microscopy (AFM) topography image). **c**, Schematic diagram of the electricity generation induced by evaporation in the biofilm and the scanning electron microscopy (SEM) image of the *R. palustris* biofilm. **d**, Spontaneous evaporation of water in the negatively charged nanochannels in the biofilm. **e**, The diffusion current ($I_{diffuse}$) and the opposing streaming current (I_{sc}) arising from ionic gradient and coulomb forces, respectively. **f**, Electroactive microorganisms uptake extracellular hydrovoltaic electrons. Data represent mean \pm SD from $n = 3$ technical replicates. Scale bars: 1 μm in **c**.

References

- R1. Xue, G. et al. Water-evaporation-induced electricity with nanostructured carbon materials. *Nat. Nanotechnol.* **12**, 317-321 (2017).
- R2. Yoon, S. G. et al. Evaporative electrical energy generation via diffusion-driven ion-electron-coupled transport in semiconducting nanoporous channel. *Nano Energy* **80**, 105522 (2021).
- R3. Zhang, Z. et al. Emerging hydrovoltaic technology. *Nat. Nanotechnol.* **13**, 1109-1119 (2018).
- R4. Wang, X. et al. Hydrovoltaic technology: from mechanism to applications. *Chem. Soc. Rev.* **51**, 4902-4927 (2022).

- R5. Zheng, C. et al. Materials for evaporation-driven hydrovoltaic technology. *Interdisciplinary Mater.* **1**, 449-470 (2022).
- R6. Wang, X. et al. Hydrovoltaic technology: from mechanism to applications. *Chem. Soc. Rev.* **51**, 4902-4927 (2022).
- R7. Yu, F. et al. High hydrovoltaic power density achieved by universal evaporating potential devices. *Adv. Sci.* **10**, 2302941 (2023).
- R8. Kong, H. et al. Mixed-dimensional van der Waals heterostructures for boosting electricity generation. *ACS nano* **17**, 18456-18469 (2023).
- R9. Swainsbury, D. J. et al. Structures of *Rhodospseudomonas palustris* RC-LH1 complexes with open or closed quinone channels. *Sci. Adv.* **7**, eabe2631 (2021).

Reviewer #3 (Remarks to the Author):

[1] *The authors have proposed an interesting and novel idea to grow electroactive microbes via hydrovoltaic energy. While the idea is commendable, the manuscript does not sufficiently address the underlying mechanisms. My specific comments to the authors are given below:*

Response: Thank you very much for your comments. We have addressed your specific comments in a point-by-point response provided below.

[2] *Line 116-117: The authors must make it clear how non-evaporation of water increases catabolic metabolism? What is the underlying reason? Similarly, how does water evaporation enhance anabolic metabolism?*

Response: Thanks so much for your professional advice. In the absence of water evaporation, the top and bottom of the biofilm did not establish an ionic concentration gradient (hydrovoltaic electric field). Therefore, under non-evaporative condition, high-density microbial cells could not access external energy to sustain their cellular metabolism, which could prompt them to utilize their own biomass (*Nat. Rev. Microbiol.* **11**, 83-94, 2013). The energy and nutrients provided by the death and lysis of a fraction of the cells could support the remaining viable fraction (*P. Natl. Acad. Sci. USA* **96**, 4023-4027, 1999; *Science* **259**, 1757-1760, 1993). This process, however, led to a decrease in microbial biomass and cellular activity, which aligned with our experimental findings (**Figs. 3a, 3b**).

Previous evidences have been revealed that an external electric field could accelerate microbial activity due to improve extracellular electron transfer and intercellular redox regulation (*Curr. Opin. Biotech.* **75**, 102687, 2022; *ISME J.* **11**, 704-714, 2017). In the presence of water evaporation, the water movement induced hydrovoltaic electric field was analogous to the externally applied bias electric field in a conventional microbial electrosynthesis that was an artificial system to verify the electron uptake process by electroactive bacteria. The M-HEG attained a hydrovoltaic electric field with a cathode potential of approximately -0.28 V vs. standard hydrogen electrode (SHE), a value that falls within the potential range necessary for *R. palustris* to accept extracellular electrons (*Sci. Adv.* **7**, eabh1852, 2021; *Nat. Commun.* **10**, 1355, 2019). Therefore, the cathode potential can drive *R. palustris* to accept extracellular electrons for intracellular carbon fixation and denitrification. As revealed by transcriptomic analyses (**Fig. 4**), the carbon fixation process and denitrification process were accelerated under the hydrovoltaic electric field. More detailed mechanisms can refer to the “Mechanisms of hydrovoltaic electron-driven microbial growth” section in main text.

Revised in text: “The high-density microbial cells could not access external energy sources to sustain their cellular metabolism, which led them to rely on their own biomass by triggering the death and lysis of a portion of the cells^{33,34}.”. Please see Page 6 Line 118-120.

References

- R1. Hoehler, T. M. & Jørgensen, B. B. Microbial life under extreme energy limitation. *Nat. Rev. Microbiol.* **11**, 83-94 (2013).
- R2. Finkel, S. E. & Kolter, R. Evolution of microbial diversity during prolonged starvation. *P. Natl. Acad. Sci. USA* **96**, 4023-4027 (1999).
- R3. Zambrano, M. M. et al. Microbial competition: *Escherichia coli* mutants that take over stationary phase cultures. *Science* **259**, 1757-1760 (1993).
- R4. Fang, Z., Tang, Y. J. & Koffas, M. A. Harnessing electrical-to-biochemical conversion for microbial synthesis. *Curr. Opin. Biotech.* **75**, 102687 (2022).
- R5. Deutzmann, J. S. & Spormann, A. M. Enhanced microbial electrosynthesis by using defined co-cultures. *ISME J.* **11**, 704-714 (2017).
- R6. Liu, X. et al. Syntrophic interspecies electron transfer drives carbon fixation and growth by *Rhodopseudomonas palustris* under dark, anoxic conditions. *Sci. Adv.* **7**, eabh1852 (2021).
- R7. Guzman, M. S. et al. Phototrophic extracellular electron uptake is linked to carbon dioxide fixation in the bacterium *Rhodopseudomonas palustris*. *Nat. Commun.* **10**, 1355 (2019).

[3] *In the setup with the biofilm and electrodes, the authors must describe how current generation is possible in the first place. Do they use a specific set of resistors to make electrons flow? Or apply a potential? It is highly unclear how electricity is generated in this setup. A clear schematic diagram along with a picture of the setup must be provided, instead of multiple images in the text and supplementary.*

Response: Thanks so much for your precious suggestion. A clear schematic diagram along with a picture of the setup was provided in **Supplementary Fig. 2**, which clearly described the configuration and components of the M-HEG system. In the M-HEG system, a biofilm was positioned between two porous titanium mesh electrodes. This sandwiched setup was then inserted into a transparent vessel and supported by the filter membrane. The top and down titanium mesh electrodes were linked to the electrical circuit through titanium wires. To enable optimal electron flow, a short circuit was created by directly connecting the top and down electrodes with titanium wires, without external resistance loading on the electrical circuit. The electrical generation performance of the M-HEG was also analyzed with various load resistances (see **Supplementary Fig. 5**), revealing a dependency of energy output on the load resistance. The M-HEG is an energy generation system driven by water evaporation, and it does not require the application of any external potential bias.

Revised in text: “This sandwiched setup was then inserted into a transparent vessel and supported by the filter membrane.”. Please see Page 13 Line 325-326.

Revised in the Supplementary information: Following figure has been modified

Supplementary Fig. 2 | Schematic and photographic illustrations of the microbial hydrovoltaic energy generation system (M-HEG). **a**, Schematic illustration of the M-HEG system. Optical microscopy image of the electrodes (top electrode and bottom electrode), scanning electron microscopy (SEM) images of the *R. palustris* biofilm and the filter membrane. **b**, Photographic illustration of the constructed M-HEG. **c**, Three-dimensional confocal laser scanning microscopy (CLSM) image of the *R. palustris*-based M-HEG biofilm at 0 day. The resulting fluorescence data was analyzed by a ZEN software and Image J software to obtain cell viability. The green and red dots represent the living cells and the dead cells, respectively, and the unit is micrometer (μm). Scale bars: 200 μm in **b** for top electrode, 2 μm in **b** for biofilms, 10 μm in **b** for filter membrane.

References

- R1. Liu, X. et al. Power generation from ambient humidity using protein nanowires. *Nature* **578**, 550-554 (2020).
 R2. Gorby, Y. A. et al. Electrically conductive bacterial nanowires produced by *Shewanella oneidensis* strain MR-1 and other microorganisms. *P. Natl. Acad. Sci. USA* **103**, 11358-11363 (2006).

[4] *It is surprising to note that the current obtained from the setup is negative (Fig 2a), while the corresponding supplemental figure 4 shows a positive voltage. Can the authors explain why it is so? Are the electrons actually flowing from the biofilm to the electrode as stated in the text? A negative current implies that the biofilm may be uptaking electrons from the electrodes. A clear explanation is needed to address this anomaly.*

Response: Thank you for the question. The obtained negative current is due to technical issues in the testing process. The M-HEG has only two electrodes (top electrode (cathode) and bottom electrode

(anode)) (**Fig. R3a** and **3b**). The bacterial cells in the biofilm would uptake electrons from the cathode electrode for cell growth and metabolism in this system. During monitoring the open-circuit voltage (V_{oc}) and short-circuit current (I_{sc}) of this system, the cathode and anode of the M-HEG was connected to the working electrode and counter/reference electrodes of an electrochemical workstation, respectively (**Fig. R3a**). We used the open-circuit voltage testing module in the electrochemical workstation to test the V_{oc} . The connection of the cathode of the M-HEG to the working electrode and the anode to the counter/reference electrodes results in a positive voltage output (**Fig. R3a**), while the opposite configuration will generate a negative voltage. It is only a technical issue encountered during the testing process.

In order to obtain the short-circuit current (I_{sc}) of M-HEG, we switched the open-circuit voltage testing module to the zero resistance ammeter (ZRA) testing module, which can test the current output of the electrochemical device itself without applying bias voltage. As shown in **Fig. R3b**, the current generated by M-HEG flowed from the cathode to the electrochemical workstation through the working electrode, and then flowed from the electrochemical workstation to the anode of M-HEG. This current direction was opposite to the default current direction of the electrochemical workstation. Hence it was a negative current. As shown in **Fig. R3c**, in a typical current test of a microbial electrosynthesis cell, the electrochemical workstation serves as a constant voltage source, providing a stable negative voltage for the microbial electro-synthesis cell, resulting in the electrochemical workstation displaying a negative current. Therefore, the currents in both cases of **Fig. R3b** and **R3c** are negative, as the currents flow towards the working electrode. Similar methods are often used for testing power generation devices such as solar cells and HEGs with other inorganic materials (*Nano Energy* **56**, 82-91, 2019; *Nano Energy* **53**, 698-705, 2018; *Small* **14**, 1704473, 2018; *ACS Appl. Mater. Interfaces* **13**, 17902-17909, 2021).

Fig. R3 Detailed display of electrical testing for M-HEG. a, Open-circuit voltage testing of the M-HEG. **b**, Short-circuit current testing of the M-HEG. **c**, Chronoamperometry testing of a typical microbial electrosynthesis.

References

R1. Lin, R. et al. Solar-powered overall water splitting system combing metal-organic frameworks derived

- bimetallic nanohybrids based electrocatalysts and one organic solar cell. *Nano Energy* **56**, 82-91 (2019).
- R2. Shao, C. et al. Wearable fiber form hygroelectric generator. *Nano Energy* **53**, 698-705 (2018).
- R3. Xu, T. et al. Electric power generation through the direct interaction of pristine graphene-oxide with water molecules. *Small* **14**, 1704473 (2018).
- R4. Li, Y. et al. Asymmetric charged conductive porous films for electricity generation from water droplets via capillary infiltrating. *ACS Appl. Mater. Interfaces* **13**, 17902-17909 (2021).

[5] *One of my concerns is that water evaporation may not provide sufficient activation energy for the biofilm to generate electricity. Could the authors show that the thermodynamics of water evaporation favours energy generation? I understand this is an additional step, but it is necessary to show that the thermodynamics actually favour the reaction pathway proposed here. We understand the thermodynamics pretty well for conventional microbial fuel cells so it must also be accounted for here.*

Response: Thank you for your suggestion. The generation of hydrovoltaic electricity by biofilms through water evaporation has been previously verified by our group and other research teams (*Sci. Adv.* **8**, eabm8047, 2022; *Nat. Commun.*, **13**, 4369, 2022; *Bioresource Technol. Rep.* **22**, 101379, 2023). In particular, our prior study validated the capability of the bacteria-based HEG system to charge a capacitor and subsequently illuminate an LED light (**Fig. R4**, *Sci. Adv.* **8**, eabm8047, 2022), confirming the feasibility of M-HEGs as potential energy sources in future.

At present, the thermodynamic calculations for the M-HEG are unavailable due to the complexity of the energy generation process and the absence of certain key calculation parameters. However, it is possible to calculate the energy conversion efficiency from water evaporation to electrical energy. The efficiency is defined as the ratio of output electric energy to input kinetic energy, as outlined in the general description of the streaming mechanism. In theory, a positive energy conversion efficiency within this distinct system should suggest that energy generation via water evaporation is thermodynamically plausible.

Liu *et al.* (*Nat. Commun.*, **13**, 4369, 2022)) have previously proposed a method for calculating energy conversion efficiency, which can be outlined in detail as follows:

1) the input kinetic power $P_{kinetic}$ can be calculated as $P_{kinetic} = Q \times \Delta p$, where Q (cm³/s) is the volume flow rate and Δp (Pa) the pressure difference across the biofilm (*Nano Lett.* **6**, 2232-2237, 2006).

2) the maximal electric energy output can be approximated as $P_{electric} = V_{oc} \times I_{sc} / 4$ (*Nature* **578**, 550-554, 2020), where V_{oc} (V) and I_{sc} (μA) are the measured open-circuit voltage and short-circuit current, respectively, factor 4 is the constant used to calculate the maximum power when the hydrovoltaic I - V curve approximates a straight line. As a result, the conversion efficiency η can be obtained as:

$$\eta = \frac{P_{electric}}{P_{kinetic}} = \frac{V_{oc} \times I_{sc}}{4 \times Q \times \Delta p} \quad (1)$$

Where the V_{oc} and I_{sc} were 0.32 V and 0.63 μA, respectively. The Q was ~0.16 cm³/s, and calculated

by weight loss rate/vapor density. The weight loss rate ($\mu\text{g/s}$) was calculated by dividing the weight loss (13.71 g) by time (50×86400 s), where 50 is the number of days and 86,400 is the number of seconds per day. The vapor density is $\sim 20 \text{ g/m}^3$ at $30 \text{ }^\circ\text{C}$ (*Poultry Sci.* **77**, 1803-1814, 1998). The vapor pressure can be determined with given temperature and relative humidity (RH) (https://www.weather.gov/epz/wxcalc_vaporpressure). The relative humidity and temperature outside the vial at the vicinity of the biofilm device were measured by a miniature sensor (TH20BL-EX, Miaoxin, China) to be $\sim 89\%$ and $28 \text{ }^\circ\text{C}$, whereas the RH and temperature inside the vial can be treated as 100% and $30 \text{ }^\circ\text{C}$, respectively. Consequently, $\Delta p = p_{100\%RH,30^\circ\text{C}} - p_{89\%RH,28^\circ\text{C}} = 463 \text{ Pa}$.

3) Substituting these obtained values into equation (1) results in an energy conversion efficiency of $\eta = 0.07\%$. Essentially, it can be indirectly inferred that the streaming potential generated by the water evaporation process can be thermodynamically converted into electrical energy.

Fig. R4 Demonstration of M-HEG as a practical power source (cited from our previous report in *Sci. Adv.* **8**, eabm8047, 2022), showing that a capacitor could be charged by five M-HEGs connected in series and subsequently lit up the LED.

References

- R1. Liu, X. et al. Microbial biofilms for electricity generation from water evaporation and power to wearables. *Nat. Commun.* **13**, 4369 (2022).
- R2. Hu, Q. et al. Water evaporation–induced electricity with *Geobacter sulfurreducens* biofilms. *Sci. Adv.* **8**, eabm8047 (2022).
- R3. Chatterjee, A., Lal, S., Manivasagam, T. G. & Batabyal, S. K. Surface charge induced bioelectricity generation from freshwater macroalgae *Pithophora*. *Bioresource Technol. Rep.* **22**, 101379 (2023).
- R4. Qin, Y. et al. Constant electricity generation in nanostructured silicon by evaporation-driven water flow. *Angew. Chem.* **132**, 10706-10712 (2020).
- R5. Liu, X. et al. Power generation from ambient humidity using protein nanowires. *Nature* **578**, 550-554 (2020).
- R6. Mitchell, M. & Kettlewell, P. Physiological stress and welfare of broiler chickens in transit: solutions not problems! *Poultry Sci.* **77**, 1803-1814 (1998).

[6] *Lines 163-167: How big is the potential difference across the biofilm? Is it sufficiently high for*

sustained generation of current over long periods of time? And can this potential difference be maintained over time?

Response: The water evaporation-induced hydrovoltaic potential difference can reach 0.29 to 0.34 V. Under ambient conditions, water evaporation is a continuous and spontaneous process. Therefore, a relatively stable electric current can be output over long periods of time. In ideal conditions, with unsaturated water vapor pressure and a stable M-HEG, this hydrovoltaic potential difference can persist stably for an extended duration of over 30 days (*Nat. Commun.* **13**, 4369, 2022). In this study, we were able to achieve continuous stable power generation for 50 days without significant decline.

References

- R1. Liu, X. et al. Microbial biofilms for electricity generation from water evaporation and power to wearables. *Nat. Commun.* **13**, 4369 (2022).
- R2. Zheng, C. et al. Materials for evaporation-driven hydrovoltaic technology. *Interdisciplinary Mater.* **1**, 449-470 (2022).

[7] Even for the non-evaporation setup where tape is used as a sealant, there is likely to be some evaporation, though it is minimised. The authors must account for this.

Response: Thanks. You are right that a weak evaporation must be still present in the non-evaporation setup. In a sealed environment, we evaluated evaporation, which amounted to 0.037 ± 0.01 g over a 50-day period. The amount of water evaporation in the non-evaporation setup was only $2.7 \pm 0.7\%$ of that in the evaporation setup, indicating successful achievement of a non-evaporation control.

Revised in text: “The system was sealed with airtight tape to prevent water-gas exchange, resulting in an evaporation loss of $2.7 \pm 0.7\%$. This setup served as the control for absent water evaporation (non-evaporation group).” Please see Page 13-14 Line 338-340.

[8] Lines 237-238: Geobacter and Shewanella exhibited no growth. Despite this, the authors claim that these microbes maintained a well-functioning metabolic state by actually uptaking hydrovoltaic electrons. How is this possible? And how are Geobacter and Shewanella able to reduce MO and NO₃⁻ in the absence of an applied potential? These statements are not satisfactory, and need more evidence and explanation.

Response: Thanks for your comments. Our study showed that hydrovoltaic electrons could be generated in the M-HEGs based on *G. sulfurreducens* and *S. oneidensis* biofilms, which could trigger the reduction reactions of MO and NO₃⁻ (electron acceptors) to complete the electron transfer processes. The specific electron transfer pathways are depicted in **Fig. R5**. The generated hydrovoltaic electrons were stored in the outer-membrane cytochrome c (e.g., *OmcB*) of *G. sulfurreducens*, and the electrons were then transferred to the terminal reductase for methyl orange (MO) degradation. Correspondingly,

the generated hydrovoltaic electrons will be transferred through the outer membrane cytochromes (*OmcA*, *MtrC*, *MtrB*, *MtrA*) of *S. oneidensis* to the inner membrane cytochrome (*CymA*). Subsequently, they are transferred to nitrate reductase enzymes (*NapA* and *NapB*) to reduce NO_3^- to NO_2^- . It is worth noting that *G. sulfurreducens* and *S. oneidensis* could take extracellular electrons generated in the M-HEGs to reduce MO and NO_3^- at the cathode of the system. *G. sulfurreducens*-based and *S. oneidensis*-based M-HEG can generate a cathodic potential of approximately -0.45 and -0.42 V vs. standard hydrogen electrode (SHE), respectively, which are sufficient to support the bioelectrochemical reduction reactions of *G. sulfurreducens* and *S. oneidensis* (*Electrochim. Acta* **53**, 2494-2500, 2008; *Commun. Biol.* **4**, 957, 2021).

G. sulfurreducens and *S. oneidensis* are typical electroactive bacteria. They are capable of not only transferring electrons produced by oxidizing organic compounds to the extracellular environment (i.e., electricity generation) (*Biosens. Bioelectron.* **146**, 111748, 2019; *Nat. Commun.* **9**, 3637, 2018), but also directly utilizing extracellular electrons such as photoelectrons and electrode electrons to drive reduction reactions such as MO and NO_3^- (*ACS Sustainable Chem. Eng.* **7**, 15427-15433, 2019; *Appl. Environ. Microb.* **84**, e00790-00718, 2018). Li *et al.* demonstrated that the highly efficient microbial nitrate reduction by *S. oneidensis* can be maintained for at least 32 days by uptaking extracellular electrode electrons (*Cell Reports Physical Science* **4**, 101433, 2023). Dumas *et al.* found that *G. sulfurreducens* can maintain high cell viability even after 15 days by uptaking extracellular in the absence of organic electron donors (*Electrochim. Acta* **53**, 2494-2500, 2008). In this study, the metabolic activity of these two strains could be sustained for 50 days by utilizing extracellular hydrovoltaic electrons. In response to the reviewers' feedback, we have added relevant explanations and references in the manuscript.

Revised in text: “*G. sulfurreducens* and *S. oneidensis* are typical electroactive bacteria known for their ability to transfer electrons generated by oxidizing organic compounds to the extracellular environment, leading to electricity generation^{47,48}. Additionally, they can directly utilize extracellular photoelectrons and electrode electrons to drive reduction reactions. Despite facing restrictions in growth due to the absence of preferred carbon sources in the medium^{49,50}, these heterotrophic electroactive microorganisms (EAMs) were able to maintain a well-functioning metabolic state by directly taking up and utilizing hydrovoltaic electrons (Fig. 5d)”. Please see Page 10 Line 246-252.

Fig. R5 Proposed mechanism of hydrovoltaic electron transfer processes in *G. sulfurreducens* and *S. oneidensis* for bio-reduction. a, Hydrovoltaic electron reduces the methyl orange (MO) in *G. sulfurreducens*. **b,** Hydrovoltaic electron reduces the NO_3^- in *S. oneidensis*. *OmcB*, *OmcA*, *MtrB*, *MtrC* and *MtrC*, outer-membrane cytochrome c; *CymA*, cytochrome c-type protein; *NapB*, periplasmic nitrate reductase, electron transfer subunit; *NapA*, periplasmic nitrate reductase.

References

- R1. Liu, X. et al. Flagella act as *Geobacter* biofilm scaffolds to stabilize biofilm and facilitate extracellular electron transfer. *Biosens. Bioelectron.* **146**, 111748 (2019).
- R2. Li, F. et al. Modular engineering to increase intracellular NAD (H^+) promotes rate of extracellular electron transfer of *Shewanella oneidensis*. *Nat. Commun.* **9**, 3637 (2018).
- R3. Huang, S. et al. Fast light-driven biodecolorization by a *Geobacter sulfurreducens*–CdS biohybrid. *ACS Sustainable Chem. Eng.* **7**, 15427-15433 (2019).
- R4. Miller, R. B. et al. Uniform and pitting corrosion of carbon steel by *Shewanella oneidensis* MR-1 under nitrate-reducing conditions. *Appl. Environ. Microb.* **84**, e00790-00718 (2018).
- R5. Lemaire, O. N., Méjean, V. & Iobbi-Nivol, C. The *Shewanella* genus: ubiquitous organisms sustaining and preserving aquatic ecosystems. *FEMS Microbiol. Rev.* **44**, 155-170 (2020).
- R6. Jing, X. et al. Anode respiration-dependent biological nitrogen fixation by *Geobacter sulfurreducens*. *Water Res.* **208**, 117860 (2022).
- R7. Li, Y. et al. Microbial electrosynthetic nitrate reduction to ammonia by reversing the typical electron transfer pathway in *Shewanella oneidensis*. *Cell Reports Physical Science* **4**, 101433 (2023).
- R8. Dumas, C., Basseguy, R. & Bergel, A. Microbial electrocatalysis with *Geobacter sulfurreducens* biofilm on stainless steel cathodes. *Electrochim. Acta* **53**, 2494-2500 (2008).
- R9. Strycharz, S. M. et al. Gene expression and deletion analysis of mechanisms for electron transfer from electrodes to *Geobacter sulfurreducens*. *Bioelectrochemistry* **80**, 142-150 (2011).

- R10. Ford, K. C. & TerAvest, M. A. The electron transport chain of *Shewanella oneidensis* MR-1 can operate bidirectionally to enable microbial electrosynthesis. *Appl. Environ. Microb.* **90**, e01387-01323 (2024).
- R11. Rowe, A. R. et al. Tracking electron uptake from a cathode into *Shewanella* cells: implications for energy acquisition from solid-substrate electron donors. *MBio* **9**, 10-1128 (2018).
- R12. Rowe, A. R. et al. Identification of a pathway for electron uptake in *Shewanella oneidensis*. *Commun. Biol.* **4**, 957 (2021).
- R13. Guo, W., Feng, J., Song, H. & Sun, J. Simultaneous bioelectricity generation and decolorization of methyl orange in a two-chambered microbial fuel cell and bacterial diversity. *Environ. Sci. Pollut. R.* **21**, 11531-11540 (2014).

Reviewer #4 (Remarks to the Author):

[1] *I have completed my review of the manuscript titled "Growth of electroactive microorganisms using hydrovoltaic energy through natural water evaporation" by Ren et al. and would like to provide my feedback and recommendations.*

Response: Thank you for taking the time to review our manuscript.

[2] *Overall, the manuscript presents an interesting study that investigates the coupling of electron generation by hydrovoltaic energy with the microbial metabolism of the anoxygenic phototrophic bacterium *Rhodospirillum rubrum*. The authors aim to determine if *R. rubrum* can utilize electrons generated from water evaporation within a hydrovoltaic device to drive its chemoautotrophic metabolism coupled with nitrate reduction. The experimental approach is appropriate, and the results are well presented and support the conclusions and claims made by the authors.*

Response: Thanks for comments.

[3] *However, I do have a concern regarding the novelty of this work. It has been previously demonstrated by the authors (Liu et al. Sci. Adv 7, eabh1852, 2021) that *R. rubrum* is capable of accepting electrons directly from solid surfaces, as well as from another microorganisms through a syntrophic relationship. In the present work, the authors have replaced the source of electrons from being the electrode of a bioelectrochemical system to a hydrovoltaic energy device. The authors suggest that a similar mechanism involving heme-containing proteins is responsible for electron uptake, as also shown in the above-cited paper. Furthermore, the integration of hydrovoltaic devices with electroactive organisms has been previously reported (e.g., Liu et al, Nat Commun 13, 4368, 2022), including by the authors themselves (e.g., Ren et al, Chem. Eng. J. 441, 135921, 2022).*

Response: Thanks for your constructive comments. The key discovery of this study is that hydrovoltaic energy generated by water evaporation in microbial biofilms can support the growth or metabolism of electroactive microorganisms. As emphasized in the abstract, this insight suggests a previously overlooked pathway that could have important implications for the survival of microorganisms in energy-limited environments, revealing a novel microbial metabolism distinct from traditional phototrophy and chemotrophy. Patterns of growth and metabolism were observed in microorganisms with the ability to uptake extracellular electrons, such as *R. rubrum*, *Thiobacillus denitrificans*, *Moorella thermoacetica*, *Shewanella oneidensis*, and *Geobacter sulfurreducens*. This suggests that WE-HE-mediated microbial survival and growth may be a common phenomenon in the microbial world.

As noted by reviewers, the generation of hydrovoltaic energy by biofilms has been reported by our group and other research teams. However, the focus of these studies has been on the mechanism and efficiency of hydrovoltaic energy in biofilm-based MEG systems, aiming to create a sustainable and

green energy generation system (Liu et al., *Nat. Commun.* 2022, 13, 4368; Hu, Q. et al., *Sci. Adv.* 2022, 8, eabm8047; Ren et al., *Chem. Eng. J.* 2022, 441, 135921). Given the prevalent occurrence of water evaporation and biofilms in natural ecological systems, further in-depth investigations are warranted into the ecological roles of hydrovoltaic energy. Previous studies, including our own, have confirmed that *R. palustris* is an electroautotrophic microorganism capable of accepting electrons directly from solid electrodes and from electroactive microorganisms (e.g., *Geobacter*) through syntrophic interspecies electron transfer (Liu et al., *Sci. Adv.* 2021, 7, eabh1852; *Nat. Commun.* 10, 1355, 2019). This process differs from traditional phototrophy and chemotrophy, which rely on light or redox reactions between chemicals. The identification of electroautotrophic microorganisms has provided novel insights into carbon biogeochemical processes like carbon fixation and denitrification. While the utilization of extracellular electrons from electrodes has been extensively studied in artificial systems, further validation is needed to confirm the prevalence of syntrophic interspecies electron transfer in natural environments. Therefore, it is imperative to explore other naturally occurring extracellular electron sources. Motivated by the aforementioned observations, we aim to investigate the feasibility of utilizing water evaporation-induced hydrovoltaic electrons (WE-HE) within microbial biofilms as an innovative energy source to sustain and enhance microbial growth and metabolism.

References:

- R1. Liu, X. et al. Microbial biofilms for electricity generation from water evaporation and power to wearables. *Nat. Commun.* **13**, 4369 (2022).
- R2. Hu, Q. et al. Water evaporation–induced electricity with *Geobacter sulfurreducens* biofilms. *Sci. Adv.* **8**, eabm8047 (2022).
- R3. Ren, G. et al. Hydrovoltaic effect of microbial films enables highly efficient and sustainable electricity generation from ambient humidity. *Chem. Eng. J.* **441**, 135921 (2022).
- R4. Liu, X. et al. Syntrophic interspecies electron transfer drives carbon fixation and growth by *Rhodopseudomonas palustris* under dark, anoxic conditions. *Sci. Adv.* **7**, eabh1852 (2021).
- R5. Guzman, M. S. et al. Phototrophic extracellular electron uptake is linked to carbon dioxide fixation in the bacterium *Rhodopseudomonas palustris*. *Nat. Commun.* **10**, 1355 (2019).
- R6. Lu, A. et al. Growth of non-phototrophic microorganisms using solar energy through mineral photocatalysis. *Nat. Commun.* **3**, 768 (2012).
- R7. Logan, B. E., Rossi, R., Ragab, A. a. & Saikaly, P. E. Electroactive microorganisms in bioelectrochemical systems. *Nat. Rev. Microbiol.* **17**, 307-319 (2019).

[4] *In light of these previous studies, I believe it is crucial for the authors to elaborate further on how the present work adds novel elements to the existing literature. While I acknowledge the excellent quality of this work, it is important to ensure that the manuscript clearly highlights its novelty compared to the previously published studies before being considered for publication in Nature Communications.*

Response: Thanks so much for your valuable advice. While microbial metabolism commonly revolves around phototrophy and chemotrophy, the untapped potential of ubiquitous hydrovoltaic energy from natural water evaporation as a viable energy source for microbial growth has not been previously reported. Our findings present compelling evidence that *water evaporation-induced hydrovoltaic electrons (hydrovoltaic energy) produced in microbial biofilms can be used to support the growth or metabolism of electrotrophic microorganisms*. In contrast to the non-bioactive M-HEG, which previously only generated hydrovoltaic energy, this study introduces a living M-HEG with high bioactivity. This living M-HEG not only absorbs environmental heat through water evaporation to generate hydrovoltaic energy but also converts this energy into chemical energy in microorganisms. Our study highlights the widespread reliance of microorganisms on hydrovoltaic energy, a phenomenon not previously documented in the scientific literatures. This research provides a novel perspective on microbial metabolism, potentially challenging conventional notions of phototrophy and chemotrophy. Our work has significant implications for fundamental scientific research, shedding light on a potentially important, yet previously underestimated, hydrovoltaic energy-driven microbial growth process through natural water evaporation. This process might be important for supporting life in energy-limited environments and contributes to Earth's energy and nutrient cycles in ways that have been underappreciated.

The novelty of this manuscript has been highlights in the main text and can be read as follows:

In the abstract section: “These insights imply a long-neglected pathway that could have significant implications for the survival of microorganisms in energy-limited environments, highlighting a new microbial metabolism distinct from traditional phototrophy and chemotrophy.”

In the introduction section: “Consequently, there is an increasing interest in exploring alternative energy sources capable of sustaining microbial life within these challenging environments.”

“These revelations open new avenues for exploring the potential interactions between ubiquitous water evaporation and microorganisms, particularly in the realm of energy conversion and metabolism. Yet, the feasibility of directly harnessing water evaporation-induced hydrovoltaic electrons (WE-HE) within microbial biofilms as a novel energy source to sustain and promote microbial growth remains a frontier to be fully explored and delineated.”

“but it represents an important energy source due to the widespread occurrence of water evaporation and its substantial energy potential. These findings unravel a novel aspect of microbial energy metabolism, bridging a crucial gap in our understanding of how microorganisms interact with natural phenomena.”

In the discussion section: “However, the ubiquitous process of water evaporation at the water-solid interface generates hydrovoltaic electrons. This phenomenon may directly or indirectly facilitate microbial metabolism and supports the growth of other microorganisms, such as heterotrophs and

chemoautotrophs. This study advances our understanding of the potential impact of water evaporation on biological processes and energy cycling in the natural world.”

Overall, the findings highlight the prospective role of hydrovoltaic energy as an alternative and sustainable energy source within microbial ecosystems, unveiling a novel mechanism elucidating how microbial growth and energy cycles operate within natural environments.

Reviewer #1 (Remarks to the Author):

None.

Reviewer #2 (Remarks to the Author):

A key issue with the manuscript is the lack of explanation regarding the electron and proton source for *R. palustris* growth. While the authors outline six points, they don't identify the specific electron donor.

Let me elaborate my point:

For cellular growth to occur, organisms require a source of several key elements:

Carbon

Electrons

Protons

Oxygen

Energy source to combine these elements

In typical autotrophic (oxygenic) organisms (cyanobacteria, algae, plants), the sources are as follows:

Carbon and Oxygen: CO₂

Electrons and Protons (with O₂ byproduct): Water

Energy Source: Light

R. palustris demonstrates remarkable metabolic flexibility, exhibiting five distinct growth modes: Photoheterotrophic [(A)]: Utilizes light and organic compounds (e.g., acetate) as an electron and energy source. Acetate provides carbon, protons, oxygen and part of the energy source. Light provides part of the energy source (Ref: Nature Biotechnology 22, 55–61, 2004)

Photoautotrophic [(B)]: Employs light as the primary energy source and obtains electrons and protons from inorganic compounds like H₂ or thiosulfate. CO₂ serves as the carbon and oxygen source. (Ref: Nature Biotechnology 22, 55–61, 2004)

Chemoheterotrophic [(C)]: Relies solely on organic compounds (e.g., acetate) for both carbon and energy. Acetate also provides electrons, protons, and oxygen.

Chemoautotrophic [(D)]: Utilizes inorganic electron donors (e.g., H₂ or thiosulfate) and CO₂ as the carbon source for growth. This mode obtains energy from the oxidation of inorganic compounds, electrons and protons from inorganic compounds like H₂ or thiosulfate. (Ref: Nature Biotechnology 22, 55–61, 2004)

Electrotrophic [(E)] (possibly Electroheterotrophic): A recently discovered mode where an electrochemical circuit provides part of the driving force, potentially alongside organic substrates like acetate for carbon, electrons, protons, and oxygen. (Ref: Electrochimica Acta 355, 136757, 2020)

The manuscript proposes the concept of "electroautotrophic growth" for *R. palustris*, where CO₂ serves as the carbon and oxygen source, and a hydrovoltaic cell provides the driving force. However, the mechanism for electron and proton acquisition remains unclear.

Are electrons and protons extracted from water? If so, the core question is whether a single hydrovoltaic cell (Nature Nanotechnology, volume 12, pages 317–321, 2017), can directly extract electrons and protons from water using the hydrovoltaic driving force. In other words, can a single hydrovoltaic cell perform water hydrolysis?

If so: prove it (therefore show oxygen evolution).

If no, where the electrons and protons are coming from? Also in this case, please provide experimental evidence (again the original question).

Reviewer #4 (Remarks to the Author):

The authors have comprehensively addressed the issues raised by the reviewers in their detailed response. They have produced an amended and improved version of the manuscript, incorporating additional experiments and revising the text.

My primary concern in the initial submission was around the perceived lack of novelty, as the authors have extensively studied the hydrovoltaic phenomenon and demonstrated that *R. palustris* can couple its metabolism with electrons derived from solid electrodes.

In response to my comment, the authors note, "The key discovery of this study is that hydrovoltaic energy generated by water evaporation in microbial biofilms can support the growth or metabolism of electrorophic microorganisms." They also recognise that, "while the utilisation of extracellular electrons from electrodes has been extensively studied in artificial systems, further validation is needed to confirm the prevalence of syntrophic interspecies electron transfer in natural environments. Therefore, it is imperative to explore other naturally occurring extracellular electron sources."

Although it's debatable whether a hydrovoltaic cell can be considered a proxy for a natural extracellular electron source in natural environments, my main contention here is the following:

- if the core objective of the research was to investigate the biochemistry and physiology of *R. palustris* when accepting electrons from an electrode, wouldn't a simpler electrochemical system comprising an anode (generating electrons, e.g., from water electrolysis) and a cathode (serving as an electron source for *R. palustris*) be more suitable?

If, on the other hand, the novelty and focus of this work is in the integration of the extracellular electron transfer metabolic feature of *R. palustris* with a hydrovoltaic system (i.e., understanding the mechanisms of electron transfer is secondary), then this should be explicitly stated throughout the manuscript. In this case, the elements of novelty highlighted by the authors in their point-by-point response should be reconsidered.

Given the following statements, it appears that the manuscript's focus is on providing a fundamental understanding of EET in *R. palustris*:

In the abstract, while referencing the growth and metabolic pathways of electron transfer at electrodes (ln 24), the authors assert, "These insights imply a neglected pathway that could have significant implications for the survival of microorganisms in energy-limited environments, highlighting a new microbial metabolism distinct from traditional phototrophy and chemotrophy."

In the introduction, when discussing the metabolic versatility of certain organisms in energy-limited environments, they state, "Consequently, there is an increasing interest in exploring alternative energy sources capable of sustaining microbial life within these challenging environments."

Based on the above, I would recommend the authors to address this critical issue and the Editor to consider these aspects before considering publication in Nat Comm.

Responses to Editor and Reviewer Comments for:

“Growth of electroautotrophic microorganisms using hydrovoltaic energy through natural water evaporation”

Reviewer #2 (Remarks to the Author):

A key issue with the manuscript is the lack of explanation regarding the electron and proton source for R. palustris growth. While the authors outline six points, they don't identify the specific electron donor. Let me elaborate my point:

For cellular growth to occur, organisms require a source of several key elements:

Carbon

Electrons

Protons

Oxygen

Energy source to combine these elements

In typical autotrophic (oxygenic) organisms (cyanobacteria, algae, plants), the sources are as follows:

Carbon and Oxygen: CO₂

Electrons and Protons (with O₂ byproduct): Water

Energy Source: Light

R. palustris demonstrates remarkable metabolic flexibility, exhibiting five distinct growth modes:

Photoheterotrophic [(A)]: Utilizes light and organic compounds (e.g., acetate) as an electron and energy source. Acetate provides carbon, protons, oxygen and part of the energy source. Light provides part of the energy source (Ref: Nature Biotechnology 22, 55–61, 2004)

Photoautotrophic [(B)]: Employs light as the primary energy source and obtains electrons and protons from inorganic compounds like H₂ or thiosulfate. CO₂ serves as the carbon and oxygen source. (Ref: Nature Biotechnology 22, 55–61, 2004)

Chemoheterotrophic [(C)]: Relies solely on organic compounds (e.g., acetate) for both carbon and energy. Acetate also provides electrons, protons, and oxygen.

Chemoautotrophic [(D)]: Utilizes inorganic electron donors (e.g., H₂ or thiosulfate) and CO₂ as the carbon source for growth. This mode obtains energy from the oxidation of inorganic compounds, electrons and protons from inorganic compounds like H₂ or thiosulfate. (Ref: Nature Biotechnology 22, 55–61, 2004)

Electrotrophic [(E)] (possibly Electroheterotrophic): A recently discovered mode where an electrochemical circuit provides part of the driving force, potentially alongside organic substrates like acetate for carbon, electrons, protons, and oxygen. (Ref: Electrochimica Acta 355, 136757, 2020)

The manuscript proposes the concept of "electroautotrophic growth" for R. palustris, where CO₂ serves as the carbon and oxygen source, and a hydrovoltaic cell provides the driving force. However, the

mechanism for electron and proton acquisition remains unclear.

Are electrons and protons extracted from water? If so, the core question is whether a single hydrovoltaic cell (Nature Nanotechnology, volume 12, pages 317–321, 2017), can directly extract electrons and protons from water using the hydrovoltaic driving force. In other words, can a single hydrovoltaic cell perform water hydrolysis?

If so: prove it (therefore show oxygen evolution).

If no, where the electrons and protons are coming from? Also in this case, please provide experimental evidence (again the original question).

Response: Thank you for your insightful comments. You have effectively highlighted the cellular growth strategies observed in traditional processes, as well as those newly proposed with electrotrophy. The generation of hydrovoltaic electrons by biofilms through water evaporation has been validated by our team and other researchers (*Sci. Adv.* 8, eabm8047, 2022; *Nat. Commun.* 13, 4369, 2022; *Bioresource Technol. Rep.* 22, 101379, 2023). Specifically, our previous research demonstrated the capability of the bacteria-based hydrovoltaic energy generation system to charge a capacitor and subsequently illuminate an light-emitting-diode (LED) lamp (*Sci. Adv.* 8, eabm8047, 2022), which strongly supported the generation of electrons in the M-HEGs. Protons are likely derived from the dissociation of water molecules, facilitated by the presence of negatively charged nanochannels in the biofilms (*Nat. Chem.* 2, 503-508, 2010; *Chem. Rev.* 120, 10298-10335, 2020; *J. Power Sources* 317, 143-152, 2016). While preliminary mechanisms have been suggested to explain the sources of electrons and protons in hydrovoltaic cells, further investigation is essential to comprehensively explore these sources in future studies.

Revised in the text: Although some initial mechanisms have been proposed to elucidate the origins of electrons and protons in biofilms through water evaporation, further research is necessary to thoroughly investigate these sources in future studies. (Please refer to P11 L281-283)

Reviewer #4 (Remarks to the Author):

The authors have comprehensively addressed the issues raised by the reviewers in their detailed response. They have produced an amended and improved version of the manuscript, incorporating additional experiments and revising the text.

*My primary concern in the initial submission was around the perceived lack of novelty, as the authors have extensively studied the hydrovoltaic phenomenon and demonstrated that *R. palustris* can couple its metabolism with electrons derived from solid electrodes.*

In response to my comment, the authors note, "The key discovery of this study is that hydrovoltaic energy generated by water evaporation in microbial biofilms can support the growth or metabolism of electrotrophic microorganisms." They also recognise that, "while the utilisation of extracellular electrons from electrodes has been extensively studied in artificial systems, further validation is needed to confirm the prevalence of syntrophic interspecies electron transfer in natural environments.

Therefore, it is imperative to explore other naturally occurring extracellular electron sources."

Although it's debatable whether a hydrovoltaic cell can be considered a proxy for a natural extracellular electron source in natural environments, my main contention here is the following:

*- if the core objective of the research was to investigate the biochemistry and physiology of *R. palustris* when accepting electrons from an electrode, wouldn't a simpler electrochemical system comprising an anode (generating electrons, e.g., from water electrolysis) and a cathode (serving as an electron source for *R. palustris*) be more suitable?*

*If, on the other hand, the novelty and focus of this work is in the integration of the extracellular electron transfer metabolic feature of *R. palustris* with a hydrovoltaic system (i.e., understanding the mechanisms of electron transfer is secondary), then this should be explicitly stated throughout the manuscript. In this case, the elements of novelty highlighted by the authors in their point-by-point response should be reconsidered.*

*Given the following statements, it appears that the manuscript's focus is on providing a fundamental understanding of EET in *R. palustris*:*

In the abstract, while referencing the growth and metabolic pathways of electron transfer at electrodes (ln 24), the authors assert, "These insights imply a neglected pathway that could have significant implications for the survival of microorganisms in energy-limited environments, highlighting a new microbial metabolism distinct from traditional phototrophy and chemotrophy."

In the introduction, when discussing the metabolic versatility of certain organisms in energy-limited environments, they state, "Consequently, there is an increasing interest in exploring alternative energy sources capable of sustaining microbial life within these challenging environments."

Based on the above, I would recommend the authors to address this critical issue and the Editor to consider these aspects before considering publication in Nat Comm.

Response: Thanks for your suggestions and comments. Microbial metabolism commonly revolves around phototrophy and chemotrophy, while the untapped potential of ubiquitous hydrovoltaic energy from natural water evaporation as a viable energy source for microbial growth has not been previously reported. Our research provides compelling evidence that hydrovoltaic energy, generated by water evaporation in microbial biofilms, can facilitate the growth of electrotrophic microorganisms. While the efficiency of energy conversion for microbial growth through hydrovoltaic energy is relatively low compared to processes like photosynthesis, we posit that it could still play a significant role in supporting microbial survival and growth in energy-constrained environments, considering the widespread presence of microbial biofilms and water evaporation conditions.

As highlighted by the reviewer, the integration of the extracellular electron uptake metabolic feature of *R. palustris* with a hydrovoltaic system is a key observation in our study. Furthermore, we have extensively explored the biochemistry and physiology of *R. palustris* in relation to accepting electrons from hydrovoltaic cells. Additionally, our investigation into potential natural electron sources from water evaporation for the electrotrophic growth of electroautotrophy holds significant environmental

implications, as these sources may serve as previously overlooked energy reservoirs in nature. This is distinct from a simple electrochemical system comprising an anode (generating electrons, e.g., from water electrolysis) and a cathode (serving as an electron source for *R. palustris*). In response to your feedback, the abstract had been revised with the assistance of the editor to emphasize the novelty and significance of this study. Furthermore, both the abstract and main text had been further updated to underscore the extracellular electron uptake properties of electroautotrophic bacteria. The revised abstract and text now read as follows:

Revised in the text: “It has been previously shown that devices based on microbial biofilms can generate hydrovoltaic energy from water evaporation. However, the potential of hydrovoltaic energy as an energy source for microbial growth has remained unexplored. Here, we show that the electroautotrophic bacterium *Rhodopseudomonas palustris* can directly utilize evaporation-induced hydrovoltaic electrons for growth within biofilms through extracellular electron uptake, with a strong reliance on carbon fixation coupled with nitrate reduction. We obtained similar results with two other electroautotrophic bacterial species. Although the energy conversion efficiency for microbial growth based on hydrovoltaic energy is low compared to other processes such as photosynthesis, we hypothesize that hydrovoltaic energy may potentially contribute to microbial survival and growth in energy-limited environments, given the ubiquity of microbial biofilms and water evaporation conditions.” (Please refer to P2 L16-25)

Revised in the text: “which not only supports the survival of microorganisms but also enables their growth through extracellular electron uptake.” (Please refer to P4 L76-77)

“it remains a plentiful and widely available energy source due to the prevalence of microbial biofilms and water evaporation conditions.” (Please refer to P11 L278-279)

“Furthermore, the study explored the potential of other microorganisms to utilize hydrovoltaic energy through extracellular electron uptake,” (Please refer to P12 L304-306)

References:

1. Hu, Q. et al. Water evaporation–induced electricity with *Geobacter sulfurreducens* biofilms. *Sci. Adv.* **8**, eabm8047 (2022).
2. Liu, X. et al. Microbial biofilms for electricity generation from water evaporation and power to wearables. *Nat. Commun.* **13**, 4369 (2022).
3. Chatterjee, A., Lal, S., Manivasagam, T. G. & Batabyal, S. K. Surface charge induced bioelectricity generation from freshwater macroalgae *Pithophora*. *Bioresource Technol. Rep.* **22**, 101379 (2023).
4. Chen, Y. et al. Enhancement of anhydrous proton transport by supramolecular nanochannels in comb polymers. *Nat. Chem.* **2**, 503-508 (2010).
5. Lynch, C. I., Rao, S. & Sansom, M. S. Water in nanopores and biological channels: A molecular simulation perspective. *Chem. Rev.* **120**, 10298-10335 (2020).
6. Singha, S. et al. Highly efficient sulfonated polybenzimidazole as a proton exchange membrane

for microbial fuel cells. *J. Power Sources* **317**, 143-152 (2016).